



# Oxidation of low-molecular weight organic compounds in cloud droplets: global impact on tropospheric oxidants

Simon Rosanka[1], Rolf Sander[2], Bruno Franco[3], Catherine Wespes[3], Andreas Wahner[1], and
Domenico Taraborrelli[1]

[1]Forschungszentrum Jülich GmbH, Institute of Energy and Climate Research, IEK-8: Troposphere, Jülich, Germany
[2]Atmospheric Chemistry Department, Max-Planck Institute of Chemistry, Mainz, Germany
[3]Université libre de Bruxelles (ULB), Spectroscopy, Quantum Chemistry and Atmospheric Remote Sensing (SQUARES),
Brussels 1050, Belgium

**Correspondence:** Simon Rosanka (s.rosanka@fz-juelich.de)

**Abstract.** In liquid cloud droplets, superoxide anion ($O_{2(aq)}^-$) is known to quickly consume ozone ($O_{3(aq)}$), which is relatively insoluble. The significance of this reaction as tropospheric $O_3$ sink is sensitive to the abundance of $O_{2(aq)}^-$ and therefore to the production of its main precursor, hydroperoxyl radical ($HO_{2(aq)}$). The aqueous-phase oxidation of oxygenated volatile organic compounds (OVOCs) is the major source of $HO_{2(aq)}$ in cloud droplets. Hence, the lack of explicit aqueous-phase chemical kinetics in global atmospheric models leads to a general underestimation of clouds as $O_3$ sinks. In this study, the importance of in-cloud OVOC oxidation for tropospheric composition is assessed by using the Chemistry As A Boxmodel Application (CAABA) and the global atmospheric model ECHAM/MESSy (EMAC), which are both capable of explicitly representing the relevant chemical transformations. For this analysis, three different in-cloud oxidation mechanisms are employed: (1) one including the basic oxidation of $SO_{2(aq)}$ via $O_{3(aq)}$ and $H_2O_{2(aq)}$, which thus represents the capabilities of most global models, (2) the more advanced standard EMAC mechanism, which includes inorganic chemistry and simplified degradation of methane oxidation products, and (3) the detailed in-cloud OVOC oxidation scheme Jülich Aqueous-phase Mechanism of Organic Chemistry (JAMOC). By using EMAC, the global impact of each mechanism is assessed focusing mainly on tropospheric volatile organic compounds (VOCs), $HO_x$ ($HO_x$=OH+HO_2$), and $O_3$. This is achieved by performing a detailed $HO_x$ and $O_3$ budget analysis in the gas- and aqueous-phase. The resulting changes are evaluated against $O_3$ and methanol ($CH_3OH$) satellite observations from the Infrared Atmospheric Sounding Interferometer (IASI) for 2015. In general, the explicit in-cloud oxidation leads to an overall reduction of predicted OVOCs levels, and reduces EMAC's overestimation of some OVOCs in the tropics. The in-cloud OVOC oxidation shifts the $HO_2$ production from the gas- to the aqueous-phase. As a result, the $O_3$ budget is perturbed with scavenging being enhanced and the gas-phase chemical losses being reduced. With the simplified in-cloud chemistry, about 13 Tg a$^{-1}$ of $O_3$ are scavenged, which increases to 336 Tg a$^{-1}$ when JAMOC is used. The highest $O_3$ reduction of 12 % is predicted in the upper troposphere/lower stratosphere (UTLS). These changes in the free troposphere significantly reduce the modelled tropospheric ozone columns, which are known to be generally overestimated by EMAC and other global atmospheric models.



## 1 Introduction

Aqueous-phase chemistry in cloud droplets differs significantly from gas-phase chemistry, mainly due to photolysis enhanced by scattering effects within cloud droplets (Bott and Zdunkowski, 1987; Mayer and Madronich, 2004), faster reaction rates, and chemical reactions that do not occur in the gas-phase (Herrmann, 2003; Epstein and Nizkorodov, 2012). Moreover, the conversion of nitrogen monoxide (NO) to nitrogen dioxide ($NO_2$) by peroxy radicals ($RO_2$) essentially does not take place in liquid droplets because NO is very insoluble. Compared to gas-phase chemistry, models of aqueous-phase chemistry still suffer from large uncertainties and most global models only include rudimentary implementations (Ervens, 2015). In general, warm (liquid) clouds can act as a sink for ozone ($O_3$) and its precursors in the troposphere. Figure 1 gives an overview of the inorganic aqueous-phase chemistry for $O_{3(aq)}$ according to the mechanism by Staehelin et al. (1984). When $O_3$ is taken up into cloud droplets, it is mainly destroyed via:

$$O_{3(aq)} + O_{2(aq)}^- \rightarrow O_{3(aq)}^- + O_{2(aq)} \tag{R1}$$

The superoxide anion $O_{2(aq)}^-$ is in equilibrium with its conjugate acid, the hydroperoxyl radical ($HO_{2(aq)}$):

$$HO_{2(aq)} \rightleftharpoons O_{2(aq)}^- + H_{(aq)}^+ \tag{R2}$$

Here, $HO_{2(aq)}$ is either scavenged from the gas phase or produced by photo-oxidation inside the cloud droplet. The realistic representation of clouds as $O_3$ sinks is thus sensitive to a proper representation of $HO_{2(aq)}$ in cloud droplets.

The importance of aqueous-phase chemistry for tropospheric $O_3$ has already been the topic of many earlier studies. Lelieveld and Crutzen (1990) proposed that clouds strongly influence $O_3$, $HO_x$ ($HO_x=HO_2+OH$), and $NO_x$ ($NO_x=NO+NO_2$). They concluded that under high-$NO_x$ conditions, the net-$O_3$ production is decreased by as much as 40 %. However, Liang and Jacob (1997) suggested that Lelieveld and Crutzen (1990) grossly overestimated the impact of clouds on $O_3$ because they made the arduous assumption that the methyl peroxy radical ($CH_3O_2$) could have the same solubility as $HO_2$. They predicted that clouds reduce tropospheric $O_3$ by less than 3 % in summer. A major aqueous-phase source of $HO_{2(aq)}$ is the oxidation of water soluble oxygenated volatile organic compounds (OVOCs). By not considering additional in-cloud $HO_{2(aq)}$ sources, Liang and Jacob (1997) underestimated $O_{2(aq)}^-$ concentrations dampening the in-cloud destruction of $O_{3(aq)}$. An extensive and explicit in-cloud OVOC oxidation scheme for global models, including the global ECHAM/MESSy Atmospheric Chemistry (EMAC) model, is currently not available. Most global models only include basic sulfur dioxide ($SO_{2(aq)}$) oxidation as the only in-cloud $O_{3(aq)}$ destruction pathway in the aqueous-phase (Ervens, 2015). By neglecting in-cloud OVOC oxidation, aqueous-phase $HO_{2(aq)}$ concentrations are very likely underestimated in EMAC and other global models. Thus, it is expected that global atmospheric models underestimate clouds as $O_3$ sinks. To date no global estimates of $O_3$ loss by scavenging has been reported.

Within this study, the global importance of in-cloud OVOC oxidation on tropospheric volatile organic compounds (VOCs), $HO_x$, and $O_3$ is addressed. For the detailed in-cloud OVOC oxidation scheme, the Jülich Aqueous-phase Mechanism of Organic Chemistry (JAMOC) suitable for global model applications is developed and implemented into the atmospheric chemistry mechanism Module Efficiently Calculating the Chemistry of the Atmosphere (MECCA) in our companion paper by Rosanka et al. (2020a). In JAMOC, the phase transfer of species containing up to ten carbon atoms is taken into account and a selection of





species containing up to four carbon atoms is considered to react in the aqueous-phase. Isoprene ($C_5H_8$), the most abundantly emitted VOC, is not explicitly dissolved in cloud droplets but many of its oxidation products explicitly react inside cloud droplets. In the aqueous phase, OVOCs mainly react with hydroxyl radicals ($OH_{(aq)}$) during daytime and with nitrate radicals ($NO_{3(aq)}$) during nighttime.

In this study, JAMOC is implemented into the global model EMAC (Sect. 2). The performance of JAMOC is compared to the performance of an aqueous-phase mechanism including only minimal aqueous-phase chemistry and of the standard mechanism of EMAC (each presented in Sect. 2.1). In order to understand the mechanistic behind the impact of in-cloud OVOC oxidation on a single air parcel, a box-model study is performed in Sect. 3. Afterwards, the impact on a global scale is analysed (Sect. 4). The analysis focuses on a selection of VOCs, $HO_x$, and $O_3$. The multiphase chemistry of JAMOC is expected to impact

tropospheric organic acids, which will be the topic of a further study. When considering the global $O_3$ budget, odd oxygen is also analysed to account for rapid cycling between species of the $O_x$ family. In the scope of this study, $O_x$ is defined as:

$$O_x \equiv O + O_3 + NO_2 + 2 \times NO_3 + 3 \times N_2O_5 + HNO_3 + HNO_4 + ClO + HOCl + ClNO_2 + ClNO_3 + BrO +$$
$$HOBr + BrNO_2 + 2 \times BrNO_3 + PANs + PNs + ANs + NPs \tag{1}$$

where PANs are peroxyacyl nitrates, PNs are alkyl peroxy nitrates, ANs are alkyl nitrates, and NPs are nitrophenols. In Sect. 4, all performed EMAC simulations are evaluated against satellite observations of $O_3$ and methanol ($CH_3OH$) obtained

from the Infrared Atmospheric Sounding Interferometer (IASI). Model uncertainties are discussed in Sect. 5, followed by a general conclusion (Sect. 6).

## 2    Modelling approach

The aqueous- and gas-phase mechanisms are presented in Sect. 2.1. They are used within two different modelling frameworks: a box-model and a global atmospheric model. The box-model, used to investigate the local impact on an air parcel, is presented

in Sect. 2.2, and the global chemical atmospheric model in Sect. 2.3. Section 2.4 provides an overview of all simulations performed in this study.

### 2.1    The chemical mechanism

The study is based on the comparison of three different aqueous-phase mechanisms (Sect. 2.1.1). While they are characterised by different complexity, especially in the species and reactions taken into account, they are all coupled to the same gas-phase

mechanism (Sect. 2.1.2).

### 2.1.1    Aqueous-phase

The first aqueous-phase mechanism includes the uptake of a few soluble compounds, their acid-base equilibria, and the oxidation of $SO_{2(aq)}$ via $O_{3(aq)}$ and $H_2O_{2(aq)}$. This mechanism was applied by Jöckel et al. (2006) and is considered to represent the capabilities of most global models (Ervens, 2015). The second aqueous-phase mechanism includes an advanced scheme,



representing more than 150 reactions (Tost et al., 2007; Jöckel et al., 2016). It includes in-cloud $HO_{x(aq)}$ chemistry and the destruction of $O_{3(aq)}$ by $O_{2(aq)}^-$, but misses a detailed in-cloud OVOC oxidation scheme. This mechanism can be considered to be the current standard mechanism used in EMAC. The last aqueous-phase mechanism is the complex OVOC oxidation scheme JAMOC developed in our companion paper by Rosanka et al. (2020a). This mechanism is based on the box-model mechanism CLoud Explicit Physico-chemical Scheme (CLEPS 1.0, Mouchel-Vallon et al. (2017)). In order to make it appli-

cable for global models, Rosanka et al. (2020a) reduced the number of species (i.e. only a selection of species containing up to four-carbon atoms), which react within the aqueous-phase. Still, the phase transfer of soluble species containing up to ten carbon atoms is represented in JAMOC. In addition to CLEPS, Rosanka et al. (2020a) extended JAMOC by: (1) simulating hydration and dehydration explicitly, (2) taking the oligomerisation of formaldehyde, glyoxal and methylglyoxal into account, (3) adding further aqueous-phase photolysis reactions, and (4) considering the gas-phase photo-oxidation of new outgassed

species. A complete description of JAMOC, including a list of all reactions, is available in Rosanka et al. (2020a). Even though Fenton's chemistry is an in-cloud source of $OH_{(aq)}$, this chemistry is not considered in this study (switched off in JAMOC), due to missing global iron (Fe) distributions and emissions in EMAC. The associated uncertainties for excluding this $OH_{(aq)}$ sources are discussed in Sect. 5.

### 2.1.2  Gas-phase

The Mainz Organic Mechanism (MOM, Sander et al., 2019) is used to model gas-phase chemistry, containing an extensive oxidation scheme for isoprene (Taraborrelli et al., 2009, 2012; Nölscher et al., 2014), monoterpenes (Hens et al., 2014), and aromatics (Cabrera-Perez et al., 2016). In addition, comprehensive reactions schemes are considered for the modelling of the chemistry of $NO_x$, $HO_x$, $CH_4$, and anthropogenic linear hydrocarbons. VOCs are oxidised by OH, $O_3$, and $NO_3$, whereas $RO_2$ reacts with $HO_2$, $NO_x$, $NO_3$, and undergoes self- and cross-reactions (Sander et al., 2019). When the complex in-cloud

OVOC oxidation scheme JAMOC is coupled to MOM, MOM is modified following the gas-phase additions as described in Rosanka et al. (2020a).

### 2.2  Chemistry box-model CAABA

Each of the three mechanisms is implemented in the Chemistry As A Boxmodel Application (CAABA, Sander et al., 2019), in order to investigate their implications on a single air-parcel under predefined atmospheric conditions. It is capable to numeri-

cally integrate the multiphase chemical mechanism as one single system of Ordinary Differential Equations (ODEs) with appropriate phase-transfer reactions (Sander, 1999; Kerkweg et al., 2007). The Kinetic Pre-Processor (KPP version 2.2.3, Sandu and Sander (2006)) is used in MECCA to integrate these ODE systems. Further, photolysis, emissions and dry deposition of chemical species, and the exchange with other air masses outside the box (entrainment) are represented in a simplified manner.

In this study, an air parcel during summer is simulated at mid-latitude with a constant temperature of 278 K and a relative

humidity of 70 %. The same initial conditions are used as proposed in Rosanka et al. (2020a, see their Table 2), but the NO emissions are neglected in this study. In order to represent a realistic atmospheric cloud event and investigate the impact of the newly developed aqueous-phase mechanism, three atmospheric conditions are modelled during the simulated day. First,





CAABA is initialised at 0 UTC and no cloud droplets are present until 12 UTC. At 12 UTC a cloud is formed with droplet radii of 20 μm and a liquid water content of $3.0 \times 10^{-1}$ g L$^{-1}$. After one hour the cloud evaporates and all species outgas. The
rest of the day is simulated using the same conditions as before the cloud event.

## 2.3   Global model EMAC

The ECHAM/MESSy Atmospheric Chemistry (EMAC) model is a numerical chemistry and climate simulation system that includes submodels describing tropospheric and middle atmospheric processes and their interaction with oceans, land, and human influences (Jöckel et al., 2010). It uses the second version of the Modular Earth Submodel System (MESSy2) to link
multi-institutional computer codes. The core atmospheric model is the 5$^{th}$ generation European Centre HAmburg general circulation Model (ECHAM5, Roeckner et al., 2003). For the present study, EMAC (ECHAM5 version 5.3.02, MESSy version 2.54.0) is used at T63L90MA resolution, i.e. with a spherical truncation of T63 (corresponding to a quadratic Gaussian grid of approximately 1.875° by 1.875° degrees in latitude and longitude) with 90 vertical hybrid pressure levels up to 0.01 hPa.

In contrast to CAABA, gas- and aqueous-phase chemistry are calculated separately. In order to model the gas-phase mech-
anism MOM in the troposphere and stratosphere, the submodel MECCA is used. The SCAVenging submodel (SCAV, Tost et al., 2006) is used to simulate the removal of trace gases and aerosol particles by clouds and precipitation. SCAV calculates the transfer of species into and out of rain and cloud droplets using the Henry's law equilibrium, acid dissociation equilibria, oxidation-reduction reactions, heterogeneous reactions on droplet surfaces, and aqueous-phase photolysis reactions (Tost et al., 2006). In this study, SCAV is used to calculate the three aqueous-phase mechanisms presented in Sect. 2.1.1. Like MECCA,
SCAV treats the aqueous-phase mechanism as an ODE system and uses KPP (version 1) to solve it. This operator splitting is necessary because the ODE systems resulting from the combination of gas-phase and in-cloud aqueous-phase mechanisms would suffer from (1) a higher stiffness due to fast acid-base equilibria and phase-transfer reactions, and (2) load imbalances on High-Performance Computing (HPC) systems due to the sparsity of clouds. In both MECCA and to some degree SCAV, tagging systems are used to calculate detailed gas- and aqueous-phase $O_x$ and $HO_x$ budgets. These systems allow to estimate
the full implications of the aqueous-phase mechanism on atmospheric chemistry. The tagging system of MECCA is more sophisticated and allows to obtain reaction rates from multiple reactions and combine them into a single tracer (Gromov et al., 2010). For the tropospheric $O_x$ budget, the gas-phase chemical production and loss, and the scavenging and wet deposition are taken into account by using MECCA and SCAV, respectively. Additionally, the dry deposition of $O_x$ and many MOM species is calculated by the submodel Dry DEPosition (DDEP, Kerkweg et al., 2006) using its default scheme.

The MESSy submodel Model of Emissions of Gases and Aerosols from Nature (MEGAN) is used to model biogenic VOC emissions (Guenther et al., 2006). Global isoprene emissions are scaled to the best estimate of Sindelarova et al. (2014), which is 595 Tg a$^{-1}$. Biomass burning emission fluxes are calculated using the MESSy submodel BIOBURN, which calculates these fluxes based on biomass burning emission factors and dry matter combustion rates. For the latter, Global Fire Assimilation System (GFAS) data are used, which are based on satellite observations of fire radiative power from the Moderate Resolution
Imaging Spectroradiometer (MODIS) satellite instruments (Kaiser et al., 2012). The biomass burning emission factors for VOCs are based on Akagi et al. (2011).





The submodel SORBIT (Jöckel et al., 2010) is used to sample the model state along sun-synchronous satellite orbits, at the time of the satellite overpass, and to compare the model outputs to satellite observations obtained from the Infrared Atmospheric Sounding Interferometer (IASI, Clerbaux et al., 2009) onboard the Metop-A (IASI-A) and Metop-B (IASI-B) satellites. In particular, the Fast Optimal Retrievals on Layers for IASI Ozone (FORLI-$O_3$, version 20151001; see Hurtmans et al. (2012), for a description of the retrievals) are used for the comparison of tropospheric $O_3$ columns. In general, when analysing tropospheric burdens and budgets, the standard EMAC tropopause definition is used. Here, the tropopause is defined in the extra tropics using potential vorticity whereas temperature lapse rates are used in the tropics (Jöckel et al., 2006). However, when comparing modelled tropospheric $O_3$ columns to IASI-FORLI measurements, the troposphere is defined as ranging from the ground to 300 hPa in order to limit the influences of the stratospheric $O_3$, but to include the altitude of maximum sensitivity of IASI in the troposphere (Wespes et al., 2017). Moreover, this allows to avoid larger errors that affect the $O_3$ retrievals in the upper troposphere/lower stratosphere (UTLS) and that result in a positive column bias (Boynard et al., 2016). The evaluation of simulation results against global observational datasets of VOC abundance can be performed for only a few species. Daily global distributions of methanol total columns are available from IASI-A and IASI-B observations, using a neural network-based retrieval approach (Franco et al., 2018). Due to the limited vertical information on methanol that is contained in the IASI spectra, only total columns have been retrieved. Since the neural network-based retrievals do not rely on scene-dependent a priori information, no averaging kernels are produced and the retrieved total columns are meant to be compared at face value with model data (see Franco et al., 2018, and references therein). For this purpose, the IASI methanol measurements have been daily averaged on the EMAC T63 spatial grid. The comparisons with IASI $O_3$ and methanol data are associated associated with some observational uncertainties. IASI retrievals are obtained in the thermal infrared range, resulting in an especially high sensitivity to clouds. Appropriate filters are applied in order to account for cloud-contaminated IASI scenes observations. These filters are based on defined cloud cover thresholds, using information from the Eumetcast operational processing system (August et al., 2012). The fractional cloud cover threshold depends on the species observed. For $O_3$ and methanol, all observations with a fractional cloud cover above 13 % (Wespes et al., 2017) and 25 % (Franco et al., 2018) have been excluded, respectively. The IASI methanol retrievals are less sensitive to the presence of residual clouds since no radiative transfer model is used, resulting in a higher threshold for methanol. Of course, it cannot be completely ruled out that individual IASI measurements are locally affected by residual clouds that passed the filtering. However, due to the huge dataset used for the seasonal averages, it is considered that such an effect is diluted and is globally negligible.

## 2.4 Simulations performed

In both modelling frameworks, multiple simulations are performed. In CAABA, the impact of each aqueous-phase mechanism on a single air-parcel is investigated. For comparison, the same day is simulated in CAABA using the same initial conditions but excluding the specific cloud event at 12 UTC. The global impact is investigated by performing a reference and two sensitivity simulations with EMAC. Global simulations without any in-cloud aqueous-phase chemistry lead to unrealistic concentrations of $O_3$ and other chemical species (Tost et al., 2007). Therefore, the reference simulation includes the minimal scavenging mechanism (in the following called Scm). The two sensitivity simulations use the standard EMAC (in the following called





ScSta) and the detailed OVOC oxidation aqueous-phase mechanism (in the following called ScJAMOC). For consistency, the same simulation names are used for the CAABA simulations. In EMAC, the years 2014 and 2015 are simulated, where 2014 is discarded as a spin-up. All simulations were performed at the Jülich Supercomputing Center with the JURECA/JUWELS clusters (Jülich Supercomputing Centre, 2018, 2019).

## 3 Box model results

Figure 2 shows the time evolution of selected gas-phase species for the different aqueous-phase mechanisms Scm, ScSta and ScJAMOC for the cloud scenario of CAABA (see Sect. 2.2). For comparison, the results of the no-cloud scenario are also shown. Both Scm and ScSta have only little impact on most of the OVOCs explicitly treated in JAMOC. For some OVOCs, the phase transfer considered in Scm and ScSta leads to reduced gas-phase concentrations during the cloud event. After the cloud evaporates, gas-phase concentrations are slightly higher compared to the no-cloud scenario, since the OVOCs transferred into the cloud droplet generally do not oxidise. Within ScSta, a subset of these OVOCs (containing one carbon atom) are oxidised leading to a slight reduction compared to Scm. In contrast, ScJAMOC efficiently removes OVOCs, leading to overall reduced OVOC concentrations. Glyoxal, one of the OVOC examples presented in Fig. 2, is completely removed from the gas-phase and quickly hydrated within the cloud droplet. The irreversible oxidation of its hydrated forms and oligomers leads to a reduction of in-cloud glyoxal concentrations. In the gas-phase, glyoxal itself is produced by the oxidation of hydrocarbons. Due to low aqueous-phase $HO_x$ concentrations during the cloud event, the oxidation of these hydrocarbons is reduced. After the cloud evaporates, the higher hydrocarbon concentrations lead to some glyoxal being produced.

Each mechanism leads to changes in most gas-phase radical concentrations. As soon as the cloud droplets form, gas-phase $HO_x$ is reduced due to the uptake of radicals and radical precursors within the first minutes. This becomes evident when inspecting the results of Scm: in this mechanism, the uptake of $HO_x$ is not taken into account. Here, the gas-phase $HO_2$ concentration is still reduced due to the uptake of a few $HO_2$ sources (e.g. formaldehyde). In the case of the other mechanisms, the uptake of $HO_x$ is explicitly considered and leads to an additional reduction in gas-phase concentrations when the cloud forms. In the case of ScJAMOC and, to some extent, of ScSta, the additional partitioning of OVOCs into the cloud droplet leads to a further decrease of gas-phase $HO_x$ concentrations. The reduction of OH is in line with other modelling studies for cloud events (Tilgner et al., 2013). When the cloud evaporates, radicals and radical sources are transferred to the gas phase. For ScJAMOC, the efficient in-cloud oxidation of radical sources induces significantly lower $HO_x$ concentrations after the cloud evaporates. The photolysis of OVOCs and their oxidation within cloud droplets cause an increase of $HO_{x(aq)}$ of about 50 %.

When the cloud forms, gas-phase $O_3$ is reduced in comparison to the no-cloud scenario because of its reactive uptake into the cloud droplet. Within Scm, $O_{3(aq)}$ only reacts with $SO_{2(aq)}$, leading to only a little reduction in gas-phase $O_3$. This reduction is more pronounced for ScSta and ScJAMOC due to additional aqueous-phase sinks and the uptake of $HO_2$ into the cloud droplet. For ScJAMOC, the reduction in $O_3$ is larger due to the additional aqueous-phase $HO_{2(aq)}$ sources from OVOC oxidation. In the gas phase, the significantly reduced $HO_2$ concentrations cause $NO_x$ to increase ($HO_2$ being the major sink of $NO_x$). However, it mostly dampens the production of $O_3$ after the cloud event.



## 4 Global impact on atmospheric composition

This section evaluates the importance of in-cloud OVOC oxidation on a global scale by focusing on VOCs (Sect. 4.1), and HO$_x$ (Sect. 4.2). The importance for tropospheric O$_3$ is discussed in Sect. 4.3.

### 4.1 Impact on tropospheric VOCs

The extensive aqueous-phase OVOC oxidation scheme JAMOC considers many VOC sinks. These significantly influence the concentrations of tropospheric VOCs. In general, VOCs can be split into primarily emitted VOCs and OVOCs mostly formed

from secondary production (e.g., oxidation of primarily emitted VOCs). The main global source of primarily emitted VOCs are biogenic processes. The largest biogenic emissions take place in the equatorial region (e.g. Amazon Basin, Central Africa) with additional emissions in the Northern (NH) and Southern Hemispheric (SH) extratropics. Isoprene, the most abundant biogenic VOC, is only slightly influenced by ScJAMOC. The yearly mean tropospheric burden increases from 204 Gg (Scm) to 213 Gg (ScJAMOC). This increase is caused by changes in OH concentrations, the main isoprene oxidant (see Sect. 4.2). Primarily

emitted VOCs are quickly oxidised in the lower troposphere, leading to low concentrations in the free troposphere. The top panel of Fig. 3 shows the zonal mean of the sum of all OVOCs that are explicitly treated in JAMOC for Scm. High OVOC concentrations are predicted in the lower troposphere and at lower latitudes, consistent with strong terrestrial biogenic emissions at the Earth surface. By the general upward transport in the equatorial region, OVOCs are transported into the free troposphere. Due to deep convection events in the same region, OVOCs are even transported into the dry tropical upper troposphere. The

lower panel of Fig. 3 shows the changes of the sum of OVOCs explicitly treated in JAMOC induced in ScJAMOC. Overall, the tropospheric OVOC burden is reduced with the largest change in the tropical free troposphere. The frequent occurrence of clouds in this region and the high OVOC concentrations lead to an efficient removal of gas-phase OVOCs. The ubiquity of clouds in the NH extratropics allows for additional removal of OVOCs from the gas-phase. These results are in line with the box-model results presented above (see Fig. 2). The efficient removal of OVOCs in warm clouds significantly affect the OVOC

levels in the dry tropical upper troposphere. Here, these OVOCs act as an important HO$_x$ source, potentially influencing the production of O$_3$ (Jaeglé et al., 2001).

Table 1 provides an overview of the annual tropospheric burden for a selection of VOCs explicitly treated in JAMOC. As shown in Fig. 3, the global burden of most VOCs is reduced due to the uptake and oxidation processes implemented in ScJAMOC. Because of the low number of VOCs containing one carbon atom treated in ScSta, changes between Scm and ScSta

are only minor. The burden of some VOCs even increases in ScSta, which is caused by reduced HO$_x$ concentrations (see Sect. 4.2). The impact in ScJAMOC differs for each VOC, with some VOCs in terms of absolute changes being efficiently removed whereas others are only slightly impacted. The varying efficiency of the VOCs removal by clouds is explained by differences in their Henry's law constants, accommodation coefficients, and aqueous-phase reactivities. The burden of methanol, the OVOC containing one carbon atom for which the highest absolute change is predicted, is reduced by about 1000 Gg. For

methyl hydroperoxide the total change is lower but the relative reduction is higher, which is due to a slightly higher solubility and overall higher reaction rate constants for the oxidation via OH$_{(aq)}$ and NO$_{3(aq)}$. Formaldehyde is reduced by about 16 %.





Even though ethanol has a Henry's law constant similar to methanol, the relative reduction is still significantly smaller, due to a slower aqueous-phase oxidation. Ethylene glycol has a slow aqueous-phase oxidation, but a very high solubility, which results in a substantial reduction of its tropospheric burden. The opposite holds for ethyl hydroperoxide, which is four times

less soluble but undergoes a fast aqueous-phase oxidation. This leads to a relative change that is similar to the one of ethylene glycol. Acetaldehyde is the only OVOC for which an enhanced burden is predicted. This is partially due to newly implemented in-cloud sources, but in particular to the aqueous-phase oxidation of methylglyoxal yielding pyruvic acid, which is a known source of acetaldehyde (Berges and Warneck, 1992).

Figure 4 shows the seasonal mean methanol column for the IASI observations. In addition, the differences of Scm vs IASI

and ScJAMOC vs Scm are shown. The highest methanol columns occur close to its major biogenic sources (e.g., Amazon Basin, boreal forests). When using Scm, EMAC underestimates methanol at mid-latitudes and overestimates it close to its main tropical biogenic sources (see center column Fig. 4). Both these model inconsistencies are caused by an incorrect spatial distribution of biogenic emissions. The submodel MEGAN, used to simulate biogenic methanol emissions (see Sect. 2.3), estimates yearly biogenic methanol emissions of $104 \, \mathrm{Tg \, a^{-1}}$, which is close to the $103 \, \mathrm{Tg \, a^{-1}}$ estimated by Millet et al. (2008,

their Table 2). However, the spatial distribution of biogenic emissions from MEGAN is different to their predictions. Compared to Millet et al. (2008), MEGAN significantly overestimates biogenic emissions in the Amazon Basin, but underestimates emissions at mid- and high-latitudes. EMAC simulates the Amazon basin too dry in the dry season (September-November, SON) and consequently too hot (Hagemann and Stacke, 2015). The biogenic emissions in MEGAN are temperature-dependent and generally higher temperatures induce higher emissions. Thus, the positive bias of surface temperatures in EMAC leads to an

overestimation in the Amazon basin. Additionally, uncertainties for all coefficients used in MEGAN, related to the emissions of methanol and primarily emitted VOCs (e.g. isoprene) further influence the incorrect emission distribution. EMAC also underestimates methanol over the oceans. In the current simulation setup, the ocean is represented to only act as a methanol sink but should be considered to be a source as well over certain oceans (e.g. over the Pacific, see Millet et al., 2008). However, EMAC models the ocean as a net sink with an uptake of about $2.1 \, \mathrm{Tg \, a^{-1}}$, which is smaller than the predicted net sink from

Millet et al. (2008) of $16 \, \mathrm{Tg \, a^{-1}}$. It is thus expected that there is an additional deficiency in the representation of the gas-phase chemistry of methanol in MOM. Still, when using ScJAMOC, the model bias for methanol is partially resolved (see right column of Fig. 4). In areas where the sources are expected to be modelled correctly (i.e. Central Africa, East Asia), the additional in-cloud OVOC oxidation leads to a reduction of methanol partially resolving the model bias in these regions. However, ScJAMOC is not able to completely resolve the model bias over the Amazon basin. The positive model bias away

from its major sources (i.e. over oceans) is reduced and partially resolved. Especially during NH autumn (SON), the strong model bias over the East Pacific and the South Atlantic Ocean is reduced. At the same time, a high overestimation for Scm is observed southeast of India over the Indian Ocean. The strong El Niño event in 2015–2016 led to droughts, draining the already dry Indonesian peatland. This drying, in combination with widespread deforestation, led to strong Indonesian fires, emitting large amounts of VOCs (Parker et al., 2016). This positive model bias is strongly reduced when in-cloud methanol oxidation

is taken into account (ScJAMOC).





To the best of our knowledge, glyoxal satellite retrievals from the Ozone Monitoring Instrument (OMI, Levelt et al., 2006) are only available up to 2014, while the TROPOspheric Monitoring Instrument (TROPOMI) started its operations in late 2017. Levelt et al. (2018) report that this is due to detector degradation and the challenging nature of glyoxal retrievals. A detailed analysis for the year 2007 is performed by Alvarado et al. (2014). Figure 5 gives the yearly mean integrated glyoxal

column for Scm and the changes introduced by ScJAMOC. In the gas-phase, glyoxal is an oxidation product of hydrocarbons. Therefore, high glyoxal concentrations are predicted by EMAC close to strong biogenic hydrocarbon sources (e.g. Amazon Basin). As found with the CAABA box model, atmospheric glyoxal levels are significantly reduced by the chemical loss in cloud droplets with ScJAMOC (see Table 1). When comparing these results to satellite retrievals from Alvarado et al. (2014, their Fig. 9), it can be concluded that the spatial distribution is reasonably well captured by Scm. However, glyoxal levels

are generally overestimated in regions where biogenic emissions dominate. The additional sink introduced in ScJAMOC leads to a significant reduction of the model bias, especially in the Amazon Basin and over Central Africa. However, the model bias is not fully resolved yet in the Amazon Basin. Here, the too high biogenic hydrocarbon emissions from MEGAN are the cause of an overestimated production of glyoxal. It is important to keep in mind that the comparability with these satellite retrievals is limited due to a different year simulated. It is still expected that the yearly mean spatial distribution of biogenic

emissions is comparable for both years and mainly varies in their magnitude. To conclude, when using JAMOC (ScJAMOC) the representation of methanol and glyoxal gas-phase concentrations is significantly improved within EMAC.

### 4.2   Impact on tropospheric $HO_x$

VOCs play an important role in the production and loss of OH and $HO_2$. Thus, the additional uptake of VOCs will influence the tropospheric OH budget. In the troposphere, OH is primarily produced by the reaction of $O(^1D)$ with $H_2O$. Here, the main

source of $O(^1D)$ is the photolysis of $O_3$. Figure 6 gives the zonal mean of the total OH production of Scm and the changes predicted by ScJAMOC. OH is mainly produced in the lower troposphere by both its primary and secondary sources, whereas in the upper troposphere secondary sources dominate. Table 2 gives an overview on the tropospheric gas-phase OH sources and sinks. With ScJAMOC, the gross OH formation decreases by about 7.3 % from 280.2 $\mathrm{Tmol\,a^{-1}}$ to 259.8 $\mathrm{Tmol\,a^{-1}}$. This finding is consistent with the box model results (Fig. 2). The uptake and oxidation of VOCs in the aqueous-phase reduce the

contribution of VOCs to the OH production. However, the major reduction in the OH production is caused by overall reduced tropospheric $O_3$ concentrations. Specifically, the two largest $O_3$ sinks, namely the OH production induced by $O_3$ photolysis and the reaction of $O_3$ with $HO_2$, are reduced by 8.5 %. $O_3$ has a long atmospheric lifetime, leading to a low spatial variability in the reduction in tropospheric $O_3$. However, the reduction in VOC concentrations has a high spatial variability (see Fig. 3), largely determining the spatial distribution of the reduction in the total OH formation by ScJAMOC (lower panel Fig. 6). The

removal of VOCs containing one carbon atom presents the largest contribution to the reduction. The reduction in $HO_x$ leads to an additional reduction in the destruction of OH from $HO_x$ cross-reactions ($HO_2 + OH$ and $OH + OH$). The OH budget presented in this study compares well with earlier EMAC studies by Lelieveld et al. (2016), which used the standard in-cloud EMAC mechanism (ScSta). The relative contributions of each OH source and sink in ScSta are comparable with their reported budgets. However, they report a lower tropospheric gross OH formation of 251.2 $\mathrm{Tmol\,a^{-1}}$ while using the same tropopause





definition. This difference is mainly related to different years simulated (leading to different emissions), and a lower model resolution used (T42L31, approximately 2.8 by 2.8 degrees in latitude and longitude with 31 vertical layers). Specifically, the lower number of tropospheric levels is expected to influence tropospheric budgets.

Figure 7 shows the zonal $HO_2$ production for Scm and the changes predicted in ScJAMOC. Due to the fast interconversion within the $HO_x$ family, the spatial distribution and magnitude of the $HO_2$ production are similar to the production of OH.

Table 3 gives the gas-phase $HO_2$ budget for each simulation. The $HO_2$ production changes from about 315 $Tmol\,a^{-1}$ to 290 $Tmol\,a^{-1}$ for Scm and ScJAMOC, respectively. Lower VOC concentrations lead to a reduction in the $HO_2$ production. Here, the influence of VOCs containing one carbon atom is the highest (see Table 1). Thus, VOCs become less important as a $HO_2$ sink. The highest reduction is caused by the reduced availability of $HO_2$, significantly reducing radical-radical reactions as a $HO_2$ sink.

Table 2 and 3 also provide the in-cloud budgets for $OH_{(aq)}$ and $HO_{2(aq)}$. The representation of the aqueous-phase chemistry of $OH_{(aq)}$ in clouds strongly affects the $HO_{2(aq)}$ production. The aqueous-phase budget of $OH_{(aq)}$ differs significantly between ScSta and ScJAMOC, which explicitly treat in-cloud $HO_{x(aq)}$ kinetics. ScJAMOC has the highest total $OH_{(aq)}$ production with more than 12 $Tmol\,a^{-1}$, which is about four times higher than in ScSta. The higher increase, compared to the box-model (Sect. 3), is attributed to the specific box-model scenario (Sect. 2.2 and Rosanka et al., 2020a, their Table 2). In both ScSta

and ScJAMOC, most $OH_{(aq)}$ is formed via the destruction of $O_{3(aq)}$. In ScJAMOC, the photolysis of OVOCs leads to the second highest formation of $OH_{(aq)}$. Here, OVOCs containing one carbon atom contribute the most, while most $OH_{(aq)}$ is formed from methyl hydroperoxide. Due to higher radical concentrations, the reactions of $OH_{(aq)}$ with $O_{3(aq)}$ and radical-radical reactions in ScJAMOC contribute about four times as much to the loss of $HO_{x(aq)}$ compared to ScSta. The oxidation of OVOCs is the major $OH_{(aq)}$ sink, with OVOCs containing one carbon atom contributing the most. This oxidation leads to the

most significant production of $HO_{2(aq)}$, followed by OVOC photolysis. Due to increased aqueous-phase $OH_{(aq)}$ and $H_2O_{2(aq)}$ concentrations, the oxidation of $H_2O_{2(aq)}$ increases by a factor of four in ScJAMOC. The destruction of $O_{3(aq)}$ leads to a reduction in $O_{2(aq)}^-$. This equilibrium is therefore the dominant $HO_{2(aq)}$ sink for both ScSta and ScJAMOC, since $HO_{2(aq)}$ is in equilibrium with $O_{2(aq)}^-$ (R2). In the literature no in-cloud $HO_{x(aq)}$ budget on a global scale has been presented so far. The novel in-cloud aqueous-phase budgets can thus not be compared to earlier studies.

## 4.3   Impact on tropospheric $O_3$

The efficient oxidation of OVOCs by cloud droplets leads to elevated aqueous-phase $HO_{2(aq)}$ concentrations accelerating the in-cloud $O_{3(aq)}$ destruction. This has a significant impact on tropospheric $O_3$ levels predicted by EMAC. Table 4 gives the $O_x$ budget for the three simulations. The chemical production increases for ScSta compared to Scm. Slightly elevated $NO_x$ concentrations lead to an increased contribution of methylperoxy radicals and $RO_2$ reactions with NO, compensating

the reduced production from $HO_2$. For ScJAMOC, the chemical production decreases by about 150 $Tg\,a^{-1}$ (2.6 %), mainly caused by an overall reduction in $HO_2$ (see Sect. 4.2) and in $RO_2$ radicals due to the uptake and explicit oxidation of VOCs. The chemical loss on the other hand is reduced by about 90 $Tg\,a^{-1}$ (1.7 %) and about 420 $Tg\,a^{-1}$ (8.0 %) for ScSta and ScJAMOC, respectively. This reduction is mainly attributed to an overall reduction in tropospheric levels of $O_3$ and $HO_x$. The



loss by dry deposition reduces by about 50 Tg a$^{-1}$ (5.6 %) for ScJAMOC, due to generally reduced surface O$_3$ concentrations.

The largest change in the O$_x$ budget is related to scavenging processes. O$_x$ scavenging increases from about 150 Tg a$^{-1}$ (Scm) to about 260 Tg a$^{-1}$ (73.3 %) and 480 Tg a$^{-1}$ (220.0 %) for ScSta and ScJAMOC, respectively. Here, the biggest increase occurs for O$_3$ scavenging, due to the accelerated O$_{3(aq)}$ destruction by enhanced HO$_{2(aq)}$ (R1), which in turn enhances the O$_3$ uptake. These changes in the O$_x$ budget terms lead to a reduced O$_3$ burden. Compared to the literature, the O$_3$ burden from ScJAMOC is closer to the observational estimate from satellite retrievals for the same time period of 287-311 Tg in the

60°S-60°N latitudinal band and closer to the global tropospheric burden of 324 Tg derived from the IASI-FORLI observations (Gaudel et al., 2018, their Table 5). However, it is important to take into account that different tropopause definitions are used in the extra tropics. In Gaudel et al. (2018), the tropopause definition for IASI-FORLI is the WMO tropopause altitude definition, based on the temperature lapse rate (WMO, 1957). In this study, potential vorticity is used as tropopause definition in the extra tropics (see Sect. 2.3). All three O$_x$ budgets (Table 4) compare well with a recent multi-model comparison of Young et al.

(2018, see their Fig. 3). The chemical loss and chemical production get closer to the multi-model mean of about 4500 Tg a$^{-1}$ and about 4950 Tg a$^{-1}$, respectively. The tropospheric O$_3$ burden in ScJAMOC is now lower than the multi-model mean of about 340 Tg but closer to the observational estimate from Ziemke et al. (2011). The increased Stratospheric-Tropospheric Exchange (STE) is still lower than the multi-model mean (about 540 Tg a$^{-1}$) and the observational estimate of 489 Tg a$^{-1}$ by Olsen et al. (2013). The tropospheric O$_3$ lifetime is reduced by one day, due to higher relative changes in the O$_x$ loss than

in the tropospheric O$_3$ burden.

Figure 8 gives the zonal net-O$_x$ production for Scm and the changes in ScJAMOC. In general, O$_x$ is produced where NO$_x$ concentrations are high (close to the surface and in the upper troposphere). In the free troposphere, above the planetary boundary layer (PBL), the increased destruction of O$_3$ over the ocean leads to an overall net O$_x$ loss in the zonal mean. The changes in the chemical production and in the loss of O$_x$, and the increase of scavenging lead to changes in the net-O$_x$

production in ScJAMOC. At the surface, the net O$_x$ production increases. Here, the efficient uptake of O$_3$ sink precursors overcompensates the reduction in the chemical production and leads to a reduced chemical loss. This increase mainly occurs over continental regions. In the free troposphere above the PBL, the net-O$_x$ change is reduced leading to an increased O$_x$ destruction. This is directly caused by the efficient uptake of HO$_2$, VOCs, and O$_3$ precursors in this cloud dominated region in ScJAMOC. In the tropical UTLS, VOCs are an important HO$_2$ source. The efficient removal of VOCs in the lower troposphere

reduces the total VOC mass transported into this region (see Fig. 3). The chemical production of O$_x$ is therefore reduced in the tropical UTLS, due to a limited availability in HO$_2$.

Figure 9 and 10 give the yearly mean surface concentration and the zonal mean of O$_3$ for Scm and the changes of ScJAMOC. In general, O$_3$ concentrations are higher in the NH with the highest values found over continental areas. Overall, surface O$_3$ slightly decreases for ScJAMOC with the maximum mean reduction of about 4 nmol/mol. The decrease in surface O$_3$ is

very low where the net-O$_x$ production increases. The highest reduction in O$_3$ is predicted in the UTLS, where tropospheric O$_3$ concentrations are the highest. Here, O$_3$ is reduced by more than 12 % for ScJAMOC. Even though the total lower tropospheric change is similar in both hemispheres, the relative reduction is higher in the SH (NH: about 4 %, SH: about 10 %).





Figure 11 shows the seasonal tropospheric integrated $O_3$ columns from IASI-FORLI $O_3$ retrievals. In addition, the differences of Scm with respect to IASI-FORLI and ScJAMOC with respect to Scm are shown. As explained previously, the comparison is performed here by using tropospheric $O_3$ column integrated between the Earth surface and 300 hPa (see Sect. 2.3). To meaningfully compare the model profile to the IASI observation, the non-uniform sensitivity of the IASI-FORLI retrievals to the O3 vertical distribution was accounted for by applying the averaging kernels. It provided the model vertical distribution of O3 as would be seen by IASI. For this purpose, the model profiles sampled at the place and time of the IASI overpasses (see Sect. 2.3) were first vertically interpolated to the IASI pressure levels. Then the smoothing of the model profiles to the lower vertical resolution of IASI was performed following Rodgers (2000). In order to take the specific scene of each IASI observation into account, the averaging kernels of the different observations contained in the model grid cell have all been considered to smooth the gridded model profile. The smoothed model profiles are finally averaged to derive the smoothed gridded model profile. In Scm, EMAC generally overestimates tropospheric $O_3$ in the tropics and at mid-latitudes regionally by more than 10 DU. This general overestimation is lower but consistent with an earlier EMAC study by Jöckel et al. (2016). They report an overestimation of up to 15 DU (see their Fig. 29), based on a comparison of a nudged simulation with OMI $O_3$ retrievals using EMACs standard aqueous-phase mechanism (here ScSta). These differences can be attributed to a much simplified gas-phase chemical mechanism, a lower spatial resolution (inducing artificial dilution of $NO_x$ point sources, Fiore et al., 2003), and different emission data sets. At higher latitudes, especially during the NH winter (December-February, DJF) and spring (March-May, MAM), EMAC slightly underestimates tropospheric $O_3$. In ScJAMOC, the overall modelled $O_3$ bias compared to IASI-FORLI is reduced by 1-2 DU, improving the representation of $O_3$ in EMAC. Here, due to the long lifetime of $O_3$, the reduction in tropospheric $O_3$ is not limited to the typical cloud dominated and precipitation regions. This demonstrates the importance of a proper representation of in-cloud $O_{3(aq)}$ and OVOC oxidation chemistry in global models. By not taking these processes into account, as done in most global models (Ervens, 2015), tropospheric $O_3$ is overestimated. It is expected that the bias reduction is even more pronounced for the complete troposphere (when using the standard EMAC definition, see Sect. 2.3), since the highest relative reduction in $O_3$ is predicted in the UTLS above 300 hPa (Fig. 10). Similar to methanol, Scm strongly overestimates the tropospheric $O_3$ column west of Indonesia over the Indian Ocean in the NH autumn. This overestimation is also linked to the strong Indonesian peatland fires (Parker et al., 2016). Due to the ongoing Asian monsoon, the emitted VOCs are quickly transported to higher altitudes, where they act as $O_3$ precursors. The efficient upward transport of the biomass burning tracers isocyanic acid (HNCO) and hydrogen cyanide (HCN) during the summer monsoon phase, has already been investigated in earlier EMAC simulations by Rosanka et al. (2020b). In the same region, surface $O_3$ is also substantially reduced in ScJAMOC (Fig. 9). These results indicate that soluble OVOCs are efficiently removed by clouds. As a consequence, the reactive uptake of $O_3$ is enhanced and $O_3$ production dampens. This leads to a reduction of the modelled bias for this region and period when using JAMOC.





## 5 Model uncertainties

In our companion paper (Rosanka et al., 2020a), uncertainties related to the kinetic data used in JAMOC are already discussed. The global model simulations performed in this study suffer from additional uncertainties mainly attributed to (1) the representation of VOC emissions, and (2) missing sources of key oxidants. Each uncertainty will be shortly discussed in this section.

As demonstrated for methanol (see Sect. 4.1), a satisfactory reproduction of tropospheric VOC concentrations strongly
depends on the realistic representation of VOC emissions. As pointed out earlier, the highest uncertainty is introduced by the biogenic emission submodel MEGAN. For instance, isoprene emissions are very sensitive to temperature and light. These uncertainties are not well quantified. Drought stress also affects isoprene emissions and it is estimated to reduce the emissions by 17-50 % globally (Jiang et al., 2018; Sindelarova et al., 2014). Additionally, biomass burning emissions in Indonesia are potentially underestimated. Parker et al. (2016) pointed out that in the monsoon period of 2015, a high fraction of the
Indonesian fire emissions originates from peatland, which is known to produce significantly high VOC emissions (Akagi et al., 2011). In the GFAS retrievals used for biomass burning, the dominant fire type in Indonesia is assigned to tropical forest fires with the exceptions of a few grid points. The strength of VOC emissions for the Indonesian fire period in 2015 is therefore underestimated. It is thus expected that when using JAMOC and a realistic combination of peatland and tropical forest fire types the overestimation of tropospheric $O_3$ in this region and time period will be further reduced (see Sect. 4.3 and Fig. 11).

Fenton chemistry is a major source of in-cloud $OH_{(aq)}$ (Deguillaume et al., 2004). Even though these reactions are available in JAMOC, Fenton chemistry is not taken into account in this study, due to missing global iron (Fe) distributions and emissions in EMAC. However, Scanza et al. (2018) present an approach to implement these into a global model. Realising this approach in EMAC would make Fenton chemistry feasible in the future. From the literature, no global modelling study is known that couples this $OH_{(aq)}$ source to a detailed in-cloud OVOC oxidation scheme, making it difficult to estimate its impact on a
global scale. In the highly idealised box modelling study of Mouchel-Vallon et al. (2017), most $OH_{(aq)}$ (63 %) is produced from Fenton chemistry (see their supplemental material SM5). This indicates the importance of Fenton chemistry in areas with high iron concentrations. The major source of atmospheric iron is mineral dust. Fossil fuel and biomass burning also emit some iron. Thus, iron concentrations are high close to deserts with the highest concentrations in the Sahara, Lut, Thar, and Arabian desert (Wang et al., 2015, their Fig. 6). Not considering this $OH_{(aq)}$ source catalysed by iron might lead to an underestimation
of OVOC oxidation rates in the aqueous-phase. In particular Central Africa, a region with high biogenic VOC emissions, might be influenced by Fe being transported from the Sahara. In addition, mineral dust will be transported over the tropical Atlantic to the Amazon basin. Here, the missing $OH_{(aq)}$ source could be responsible for the underestimation of in-cloud OVOC oxidation and thus the destruction of $O_{3(aq)}$.

To conclude, the impact of the in-cloud OVOC chemistry on the tropospheric composition estimated in this study, is influ-
enced by some model and observational uncertainties. However, the findings of the simulations performed in this study are still consistent with earlier studies and improve the representation of a selection of OVOCs and the EMAC bias towards high $O_3$ concentrations. Due to their complexity, reducing the model uncertainties introduced by biogenic and biomass burning





emissions, and missing aqueous-phase Fenton chemistry, is outside the scope of this study. Model representation of the latter is expected to substantially increase the oxidation rate of OVOCs in the cloud droplets and aerosols. Additional global modelling
studies need to be performed to address these issues.

## 6  Conclusions

In this study, the influence of in-cloud oxidation of soluble OVOCs on the tropospheric gas-phase composition was studied. This was achieved by implementing the extensive aqueous-phase OVOC oxidation scheme JAMOC, initially presented by Rosanka et al. (2020a), into the global model EMAC. The mechanism considers a selection of VOCs containing up to four
carbon atoms, their acid/base and/or hydration/dehydration equilibria, and their reactions with $OH_{(aq)}$, $NO_{3(aq)}$, and other oxidants (if available). Additionally, the phase transfer of species containing up to ten carbon atoms is taken into account. In addition to the EMAC simulations, a representative cloud droplet was simulated in the box-model CAABA in order to understand all processes involved.

When in-cloud OVOC oxidation is taken into account, VOCs are efficiently removed from the gas-phase leading to generally
reduced tropospheric VOC burdens. The reduction in modelled methanol and glyoxal concentrations is in line with satellite retrievals. The overall reduction in VOC concentrations leads to lower formation rates of $HO_x$ in the gas-phase. Higher in-cloud $HO_{2(aq)}$ concentrations, formed from OVOC oxidation, lead to an accelerated destruction of $O_{3(aq)}$ in clouds. In addition, the chemical production and loss of $O_3$ in the gas-phase are reduced due to lower VOC and $HO_x$ concentrations. This results in a reduced $O_3$ burden and decreases EMAC's bias towards too high $O_3$ concentrations. In ScJAMOC, many secondary
organic aerosol (SOA) precursors are explicitly treated, impacting the formation of SOA (Blando and Turpin, 2000; Ervens et al., 2011; Ervens, 2015). The potentially enhanced SOA formation will further influence tropospheric $HO_x$ chemistry and $NO_2$ photolysis, resulting in a higher reduction of tropospheric $O_3$ and EMAC's $O_3$ bias. However, studying the influence of in-cloud OVOC oxidation on SOA formation is outside the scope of this study.

The findings in this study demonstrate the importance of in-cloud chemistry on tropospheric $O_3$. Most atmospheric global
models do not take detailed aqueous-phase chemistry into account (Ervens, 2015). With the minimal oxidation of $SO_{2(aq)}$ by $O_{3(aq)}$, which is representative for most global models, only about 13 Tg a$^{-1}$ of $O_3$ are scavenged by clouds. With explicit in-cloud OVOC oxidation considered, $O_3$ scavenging increases to about 336 Tg a$^{-1}$. This estimate neglects the $O_3$ sink in deliquescent aerosols, which might turn out to be significant as well. The predicted $O_3$ loss by clouds is significantly higher than the results suggested by Liang and Jacob (1997). Even though regional changes might be in the order of the results
presented by Lelieveld and Crutzen (1990), the global reduction does not get close to their reduction of 40 %. To conclude, global models, which neglect explicit in-cloud OVOC oxidation, significantly underestimate clouds as $O_3$ sinks and show a general tendency to overestimate tropospheric $O_3$.



*Data availability.* The simulation results are archived at the Jülich Supercomputing Centre (JSC) and are available on request. The IASI
O$_3$ data processed with FORLI-O$_3$ v0151001 can be downloaded from the Aeris portal at http://iasi.aeris-data.fr/O3/ (last access: 2 August
2020). The IASI methanol columns retrieved with the ANNI framework are available upon request.

*Author contributions.* SR and DT designed the study. SR performed the simulations and analysed the data with contributions from DT. BF
and CW acted as IASI data providers and analysts. SR and DT discussed the results with contributions from RS, BF, and AW. The manuscript
was prepared by SR with the help of all co-authors.

*Competing interests.* The authors declare that they have no competing of interest.

*Acknowledgements.* The work described in this paper has received funding from the Initiative and Networking Fund of the Helmholtz
Association through the project "Advanced Earth System Modelling Capacity (ESM)". The content of this paper is the sole responsibility of
the author(s) and it does not represent the opinion of the Helmholtz Association, and the Helmholtz Association is not responsible for any
use that might be made of the information contained. The authors gratefully acknowledge the Earth System Modelling Project (ESM) for
funding this work by providing computing time on the ESM partition of the supercomputer JUWELS at the Jülich Supercomputing Centre
(JSC). The authors gratefully acknowledge the computing time granted through JARA on the supercomputer JURECA at Forschungszentrum
Jülich. IASI is a joint mission of EUMETSAT and the Centre National d'Etudes Spatiales (CNES, France). The authors acknowledge the
AERIS data infrastructure for providing access to the IASI data in this study and ULB-LATMOS, in particular Daniel Hurtmans, for the
development of the retrieval algorithms. The research in Belgium is funded by the Belgian State Federal Office for Scientific, Technical and
Cultural Affairs, and the European Space Agency (ESA Prodex IASI Flow and B-AC SAF).





**Table 1.** Mean gas-phase tropospheric burden in 2015 for a selection of VOCs for Scm and the changes induced by ScSta and ScJAMOC. Burden values are given in Gg.

|  | Scm | $\Delta$ ScSta | $\Delta$ ScJAMOC |
|---|---|---|---|
| **$C_1$-VOCs** |  |  |  |
| Formaldehyde | 1212.3 | - 46.6 | - 204.2 |
| Methanol | 3279.3 | - 341.0 | - 998.8 |
| Methyl hydroperoxide | 1914.5 | - 32.9 | - 849.9 |
| Hydroxymethyl hydroperoxide | 67.8 | + 0.2 | - 16.0 |
| **$C_2$-VOCs** |  |  |  |
| Ethanol | 110.9 | + 0.4 | - 16.6 |
| Ethylene glycol | 3.1 | + 0.1 | - 1.4 |
| Acetaldehyde | 147.1 | + 1.7 | + 12.1 |
| Glycolaldehyde | 278.8 | - 0.9 | - 101.2 |
| Glyoxal | 44.6 | 0.0 | - 12.7 |
| Ethyl hydroperoxide | 62.9 | - 0.9 | - 28.3 |
| **$C_3$-VOCs** |  |  |  |
| Methylglyoxal | 181.8 | - 0.6 | - 35.3 |
| Isopropyl hydro peroxide | 13.0 | - 0.2 | - 4.6 |





**Table 2.** Global tropospheric mean gas- and aqueous-phase source and sink fluxes of OH for Scm and the changes induced by ScSta and ScJAMOC. All values are given in $\mathrm{Tmol\,a^{-1}}$. The aqueous-phase budget is only based on cloud droplets. Rain droplets are not taken into account.

|  | Scm | $\Delta$ ScSta | $\Delta$ ScJAMOC |
|---|---|---|---|
| **Gas-phase sources** | | | |
| $O(^1D) + H_2O$ | 96.67 | - 1.39 | - 7.11 |
| $NO + HO_2$ | 84.53 | - 0.25 | - 2.10 |
| $O_3 + HO_2$ | 32.36 | - 0.95 | - 3.93 |
| $H_2O_2 + h\nu$ | 26.70 | - 0.85 | - 1.39 |
| OVOCs | 30.40 | - 0.30 | - 5.82 |
| Other | 9.54 | + 0.01 | - 0.02 |
| Total | 280.20 | - 3.73 | - 20.37 |
| **Gas-phase sinks** | | | |
| $OH + HO_{y_g}$[a] | 49.88 | + 0.06 | - 1.90 |
| $OH + NO_y$[b] | 4.73 | + 0.01 | + 0.11 |
| $OH + CH_4$ | 32.85 | - 0.02 | - 0.35 |
| $OH + C_1$[c] | 150.90 | - 2.73 | - 16.20 |
| $OH + C_n\text{-VOCs}$ | 39.75 | - 0.15 | - 2.70 |
| Other | 2.09 | 0.00 | 0.00 |
| Total | 280.20 | - 3.73 | - 20.37 |
| **Aqueous-phase sources** | | | |
| $O_3 + O_2^-$ | - | + 1.94 | + 6.30 |
| $H_2O_2 + h\nu$ | - | + 0.95 | + 1.08 |
| $C_1\text{-VOCs} + h\nu$ | - | - | + 4.71 |
| $C_n\text{-VOCs} + h\nu$ | - | - | + 0.32 |
| Other | - | + 0.02 | + 0.02 |
| Total | - | + 2.91 | + 12.43 |
| **Aqueous-phase sinks** | | | |
| $OH + HO_{y_{aq}}$[d] | - | + 0.42 | + 2.20 |
| $C_1\text{-VOCs}$ | - | + 2.40 | + 8.98 |
| $C_n\text{-VOCs}$ | - | - | + 0.91 |
| Other | - | + 0.09 | + 0.34 |
| Total | - | + 2.91 | + 12.43 |

[a] $HO_{y_g} \equiv H_2, O_3, H_2O_2$, radical–radical reactions
[b] $NO_y \equiv NO, NO_2, HNO_2, HNO_3, HNO_4, NH_3$, N-reaction products
[c] $C_1 \equiv CO$, VOCs with one C-atom
[d] $HO_{y_{aq}} \equiv O_2^-, H_2O_2$, radical–radical reactions





**Table 3.** Global tropospheric mean gas- and aqueous-phase source and sink fluxes of $HO_2$ for Scm and the changes induced by ScSta and ScJAMOC. All values are given in $\mathrm{Tmol\,a^{-1}}$. The aqueous-phase budget is only based on cloud droplets. Rain droplets are not taken into account.

|  | Scm | $\Delta$ ScSta | $\Delta$ ScJAMOC |
|---|---|---|---|
| **Gas-phase sources** | | | |
| $OH + O_3$ | 12.51 | - 0.18 | - 0.71 |
| $H_2O_2 + OH$ | 13.86 | - 0.44 | - 0.56 |
| $HNO_4$[a] | 26.38 | - 0.52 | - 1.59 |
| $C_1$-VOCs | 214.71 | - 3.58 | - 17.76 |
| $C_n$-VOCs | 22.33 | + 0.01 | - 0.64 |
| Photolysis | 24.64 | - 0.47 | - 3.88 |
| Other | 1.26 | - 0.01 | - 0.01 |
| Total | 315.69 | - 5.19 | - 25.15 |
| **Gas-phase sinks** | | | |
| $HO_2 + O_3$ | 32.36 | - 0.95 | - 3.93 |
| $HO_2 + OH$ | 12.86 | - 0.21 | - 0.69 |
| $HO_2 + HO_2$ | 77.34 | - 2.33 | - 8.37 |
| $HO_2 + NO$ | 84.53 | - 0.25 | - 2.10 |
| $HO_2 + NO_2$ & $NO_3$ | 27.31 | - 0.44 | - 1.58 |
| $C_1$-VOCs $+HO_2$ | 47.63 | - 1.34 | - 6.74 |
| $C_n$-VOCs $+HO_2$ | 26.85 | - 0.22 | - 2.08 |
| Other | 6.81 | + 0.55 | + 0.34 |
| Total | 315.69 | - 5.19 | - 25.15 |
| **Aqueous-phase sources** | | | |
| Mass transfer | - | + 0.60 | + 0.51 |
| $H_2O_2 + OH$ | - | + 0.38 | + 1.61 |
| $C_1$-VOCs | - | + 2.39 | + 10.80 |
| $C_2$-VOCs | - | - | + 0.92 |
| Other | - | + 0.01 | + 0.09 |
| Total | - | + 3.38 | + 13.93 |
| **Aqueous-phase sinks** | | | |
| $HO_2 \rightleftharpoons O_2^- + H^+$ | - | + 2.68 | + 8.69 |
| $HO_2 + HO_{y_{aq}}$[b] | - | + 0.69 | + 5.22 |
| Other | - | + 0.01 | + 0.02 |
| Total | - | + 3.38 | + 13.93 |

[a] $HNO_4 \rightarrow NO_2 + HO_2$

[b] $HO_{y_{aq}} \equiv O_2^-$, radical–radical reactions





**Table 4.** Detailed tropospheric $O_x$ budget for Scm and the changes induced by ScSta and ScJAMOC. The gross terms as well as the relative contributions of the major contributors are given.

|  | Scm | $\Delta$ ScSta | $\Delta$ ScJAMOC |
|---|---|---|---|
| **Sources [Tg $a^{-1}$]** |  |  |  |
| Chemical production | 5895.6 | + 7.1 | - 155.8 |
| $HO_2 + NO$ | 4050.3 | - 12.8 | - 101.3 |
| $CH_3O_2 + NO$ | 1084.8 | + 13.1 | - 22.9 |
| $RO_2 + NO$ | 731.1 | + 6.7 | - 30.8 |
| Other | 29.4 | + 0.1 | + 0.1 |
| STE [a] | 355.2 | + 5.6 | + 15.3 |
| **Sinks [Tg $a^{-1}$]** |  |  |  |
| Chemical loss | 5254.7 | - 91.2 | - 423.2 |
| $O(^1D) + H_2O$ | 2317.3 | - 35.0 | - 167.3 |
| $HO_2 + O_3$ | 1550.1 | - 42.4 | - 187.6 |
| $OH + O_3$ | 599.0 | - 1.4 | - 0.6 |
| $HOBr + hv$ | 341.6 | - 0.8 | - 54.6 |
| $PhO + O_3$ [b] | 215.4 | + 1.5 | - 31.8 |
| Other | 231.3 | - 4.1 | - 81.5 |
| Dry deposition | 846.5 | - 9.1 | - 47.3 |
| $O_3$ | 801.6 | - 9.4 | - 47.1 |
| Other | 44.9 | + 0.3 | - 0.2 |
| Scavenging | 149.7 | + 112.9 | + 329.7 |
| $O_3$ | 13.2 | + 104.4 | + 323.1 |
| $N_2O_5$ | 25.0 | - 2.3 | - 2.7 |
| $HNO_3$ | 111.5 | - 0.3 | - 1.0 |
| Other | - | + 11.2 | + 10.3 |
| $O_3$ **burden [Tg]** | 348.2 | - 5.0 | - 25.0 |
| $O_3$ **lifetime [days]** | 20.3 | - 0.3 | - 1.0 |

[a] Stratospheric-Tropospheric Exchange

[b] $O_3$ loss due reaction with phenoxy radicals from oxidation of aromatics
(Taraborrelli et al., 2020)





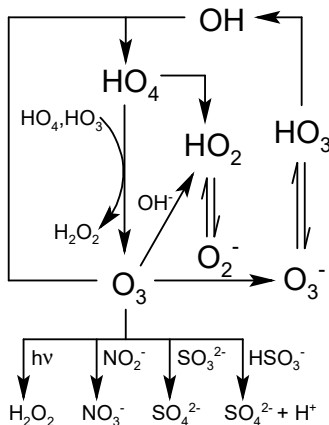

**Figure 1.** Graphical representation of inorganic aqueous-phase ozone chemistry based on Staehelin et al. (1984).

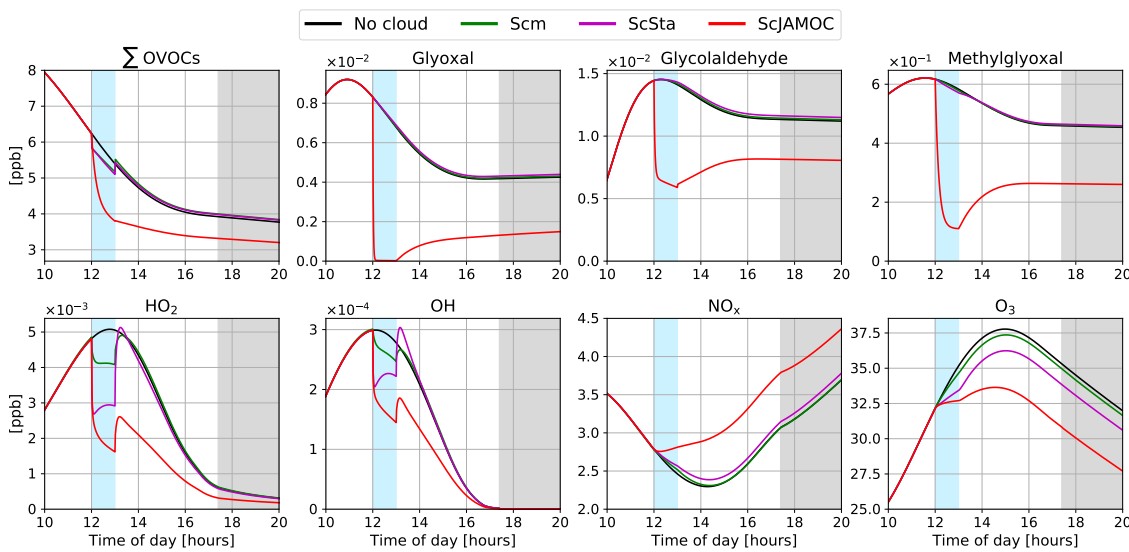

**Figure 2.** Time evolution for gas-phase mixing ratios of the sum of all the OVOCs explicitly reacting in JAMOC ($\sum$ VOCs = methanol + formaldehyde + methyl hydroperoxide + hydroxymethylhydroperoxide + ethanol + ethylene glycol + acetaldehyde + glycolaldehyde + glyoxal + hydroperoxide + methylglyoxal + isopropanol + isopropyl hydro peroxide + methacrolein + methyl vinyl ketone), glyoxal, glycolaldehyde, methylglyoxal, HO$_2$, OH, NO$_x$, and O$_3$ within the boxmodel CAABA. The time when the cloud is present (between 12 and 13 UTC) is indicated by a blue background shading. Nighttime is indicated by a grey background shading. Mixing ratios are provided for no cloud event (black line), Scm (green line), ScSta (purple line) and ScJAMOC (red line). Note that lines may overlap.

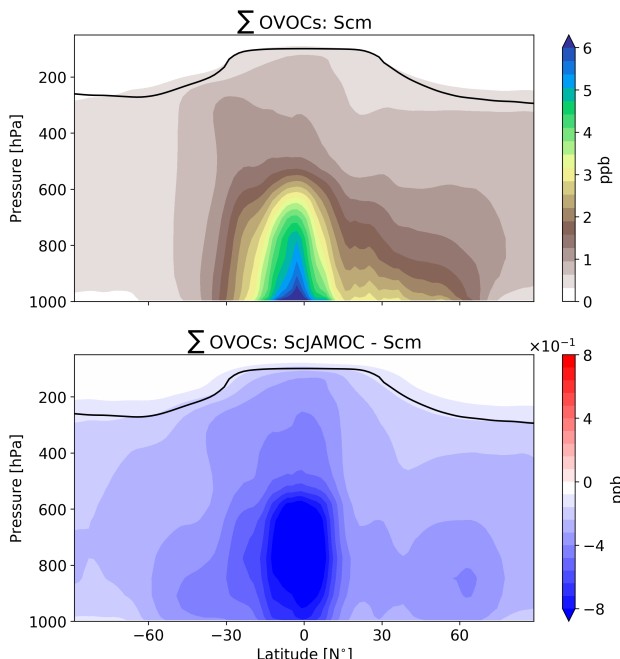

**Figure 3.** Yearly zonal mean of the sum of all the OVOCs explicitly reacting in JAMOC ($\sum$ VOCs = methanol + formaldehyde + methyl hydroperoxide + hydroxymethylhydroperoxide + ethanol + ethylene glycol + acetaldehyde + glycolaldehyde + glyoxal + hydroperoxide + methylglyoxal + isopropanol + isopropyl hydro peroxide + methacrolein + methyl vinyl ketone): for Scm (top) and in comparison to ScJAMOC (bottom). The yearly mean tropopause is depicted by a black line.



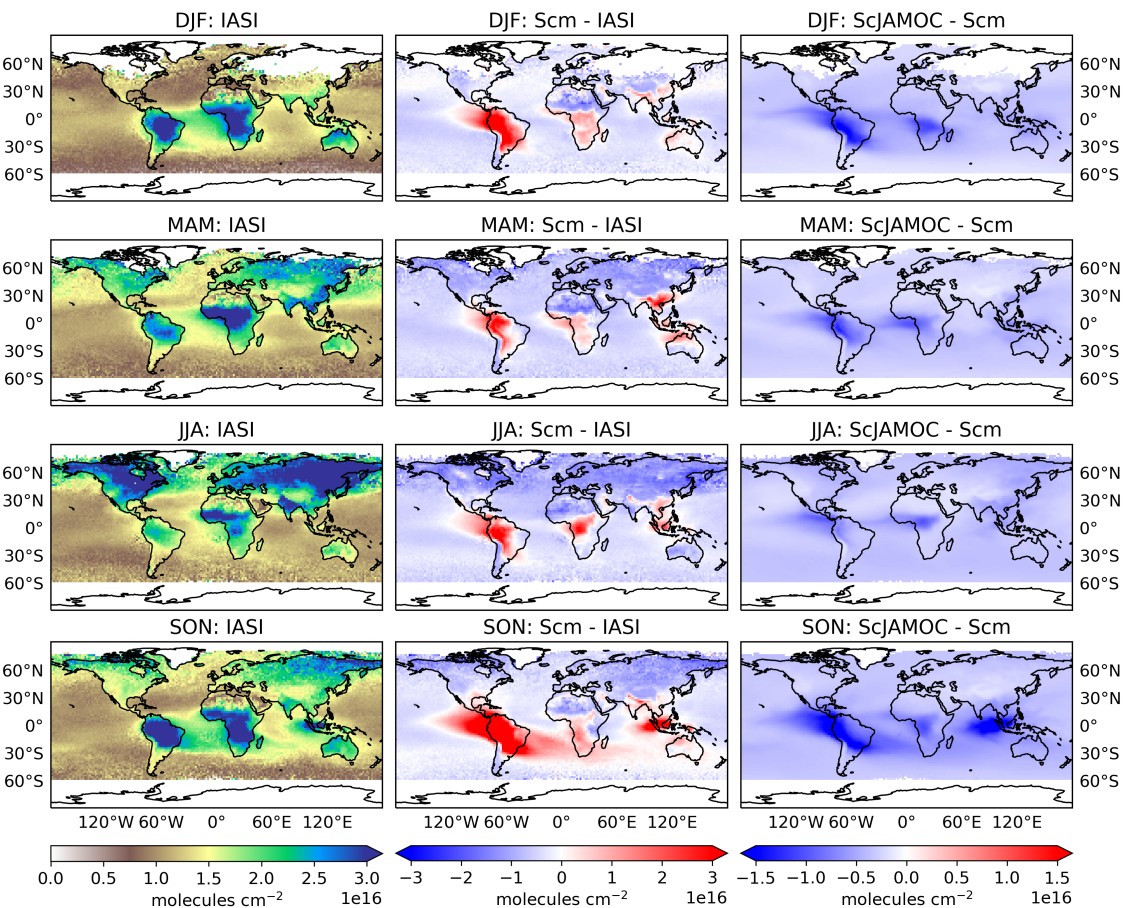

**Figure 4.** Seasonal (December-February, DJF; March-May, MAM; June-August, JJA; September-November, SON) mean integrated methanol column obtained from IASI satellite observations (left), of the Scm simulation in comparison to IASI observations (center), and of ScJAMOC in comparison to Scm (right).





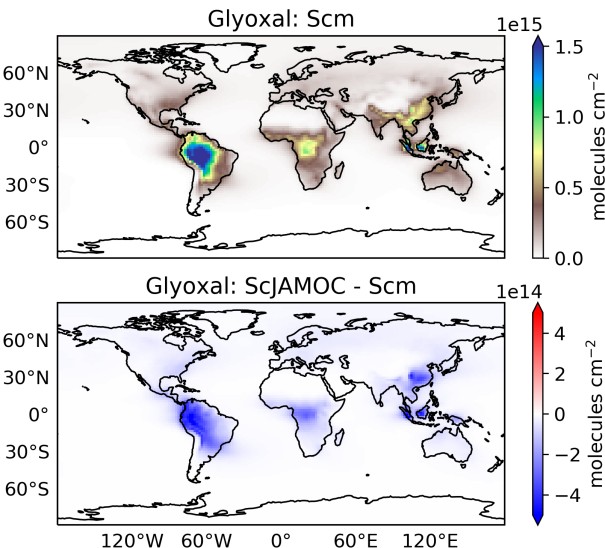

**Figure 5.** Mean integrated tropospheric glyoxal column for Scm (top) and in comparison to ScJAMOC (bottom).

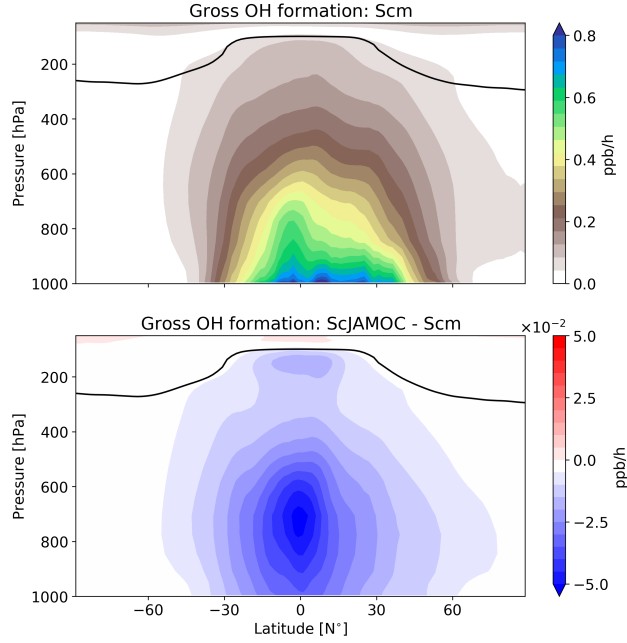

**Figure 6.** Zonal mean gross OH formation for Scm (top) and in comparison to ScJAMOC (bottom). The yearly mean tropopause is depicted by a black line.



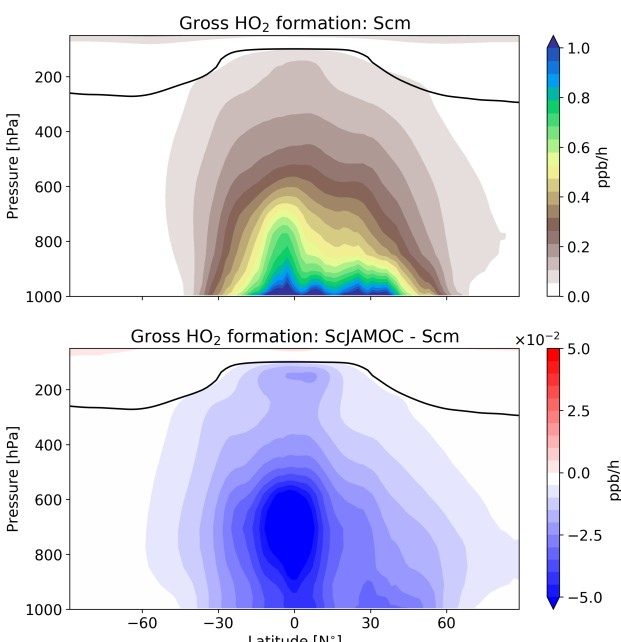

**Figure 7.** Zonal mean gross $HO_2$ formation for Scm (top) and in comparison to ScJAMOC (bottom). The yearly mean tropopause is depicted by a black line.





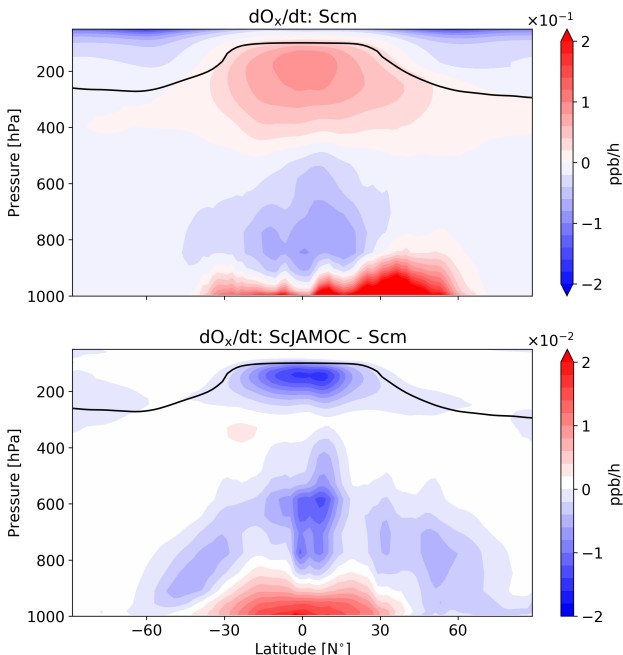

**Figure 8.** Mean zonal net-$O_x$ change for Scm (top) and in comparison to ScJAMOC (bottom). The yearly mean tropopause is depicted by a black line. Deposition in the lowest model layer is not taken into account.

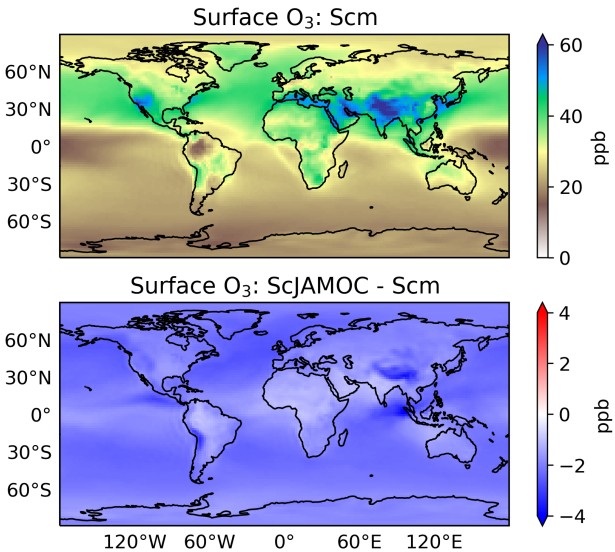

**Figure 9.** Mean surface $O_3$ concentration for Scm (top) and in comparison to ScJAMOC (bottom).

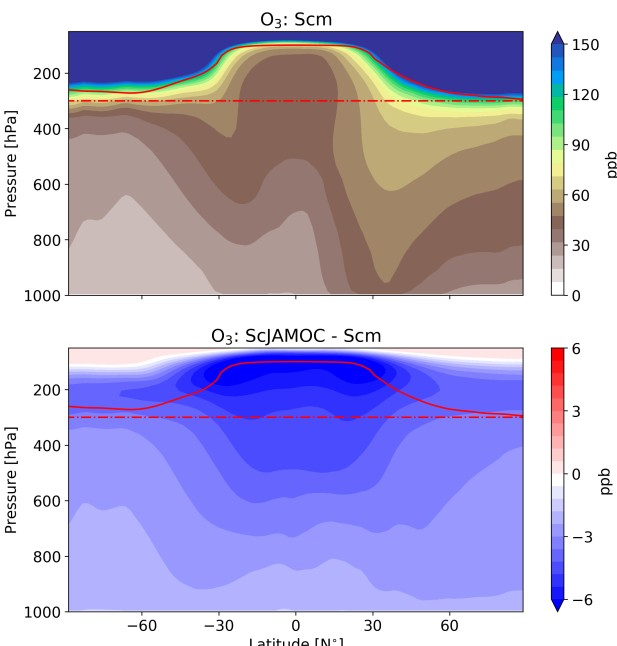

**Figure 10.** Mean zonal $O_3$ concentration for Scm (top) and in comparison to ScJAMOC (bottom). The yearly mean tropopause is depicted by a red solid line. In addition, the 300 hPa tropopause layer used for the $O_3$ IASI-FORLI comparison (see Fig. 11) is depicted by a red dash-dotted line.

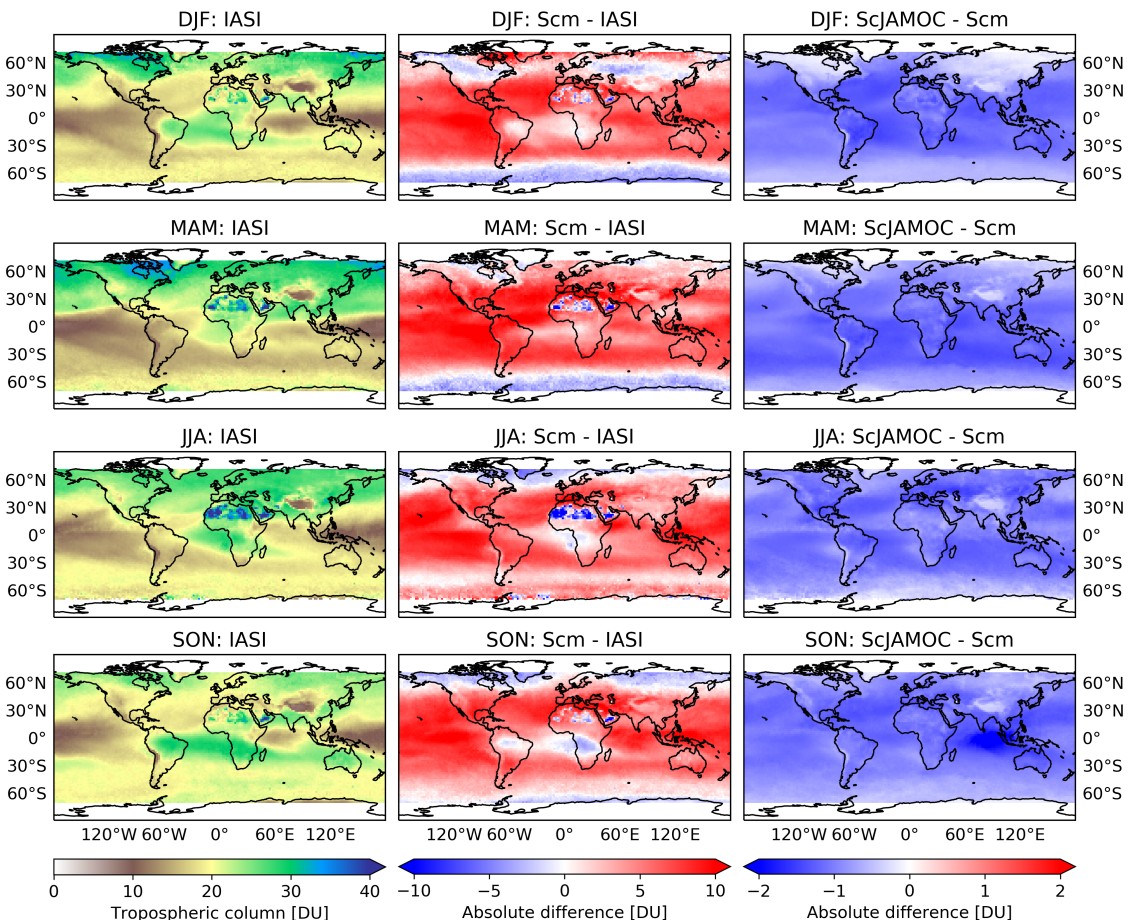

**Figure 11.** Seasonal (December-February, DJF; March-May, MAM; June-August, JJA; September-November, SON) tropospheric O$_3$ column comparison between IASI-FORLI satellite observations and EMAC. IASI-FORLI satellite observations (left), Scm simulation in comparison to IASI-FORLI observations (center), and ScJAMOC in comparison to Scm (right). For this comparison, the tropopause is defined at 300 hPa.





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
