# Peer review of "Oxidation of low-molecular weight organic compounds in cloud droplets: global impact on tropospheric oxidants"

_Atmospheric Chemistry and Physics, 2020_

## Referee Comment (RC1) · Anonymous Referee #1 · 23 Dec 2020

The manuscript presents a revisit to the aqueous chemistry of the superoxide anion in the atmosphere, using both a box and a global model. The focus of the study is on the impact of this newly implemented chemistry on VOCs, OVOCs, HOx, and ozone. Comparisons of the model against satellite methanol and ozone observations are shown, with updates decreasing EMAC's positive ozone bias. Overall the authors clearly demonstrate the importance of including this chemistry in global chemistry models. The science is generally presented in a clear and appropriate way and the manuscript as whole fits the remit of ACP. I would encourage publication effectively as is.

[Figure]

—- Specific comments —

Line 44 - A suggestion for flow of reading: Add "However" to the beginning of the sentence below.

"By not considering additional in-cloud HO2(aq) sources, Liang and Jacob (1997) underestimated O−2(aq) concentrations dampening the in-cloud destruction of O3(aq)."

Line ∼175 — Please add a table to section 2.4 to make it easier for the reader to quickly decode the simulation acronyms used elsewhere in the text.

Figure 2 caption — Consider moving expansions of families (e.g. VOCs) here and elsewhere to a table in the supplement to make the text more readable.

Table 4 - Please add some reference numbers from a multi-model study such as TOAR as a column to Table 4. This enables the reader to put these numbers in context (e.g. Loss via bromine seems quite high in this model).

Table 3 and 2 - As with Table 4, is it possible to provide some context for the numbers to another model study? Few will know where these numbers are high or low without context.

Please expand all abbreviations/acronyms in table/figure captions or at least link to a table of these (e.g. "Scm" in Table 1).

———————————————

---

## Short Comment (SC1) · 30 Dec 2020

Compliments for the excellent article that represents a major step forward in the discussion of cloud chemical effects on tropospheric composition. The development of the JAMOC scheme, accounting for comprehensive VOC chemistry, and the successful implementation in the EMAC model is an important accomplishment. The use of JAMOC brings the model significantly closer to observations of VOCs and ozone. Impressive. The results on VOCs and OVOCs, notably of aqueous phase chemistry and considering that most clouds evaporate rather than precipitate, will also offer new angles of approach in studies of organic aerosols.

[Figure]

It should be mentioned that this work was possible as it could build on the EMAC modelling framework, being the effort of a team (of which I am happy to be a member). It has set the stage for comprehensive, global atmospheric chemistry modelling, including the explicit and comprehensive account of VOCs and multiphase processes (e.g. Tost et al., 2007; Taraborrelli et al, 2009; Sander et al., 2011, 2019, Jöckel et al., 2010). I hope the article will be accepted for publication in ACP, while having a few minor comments in view of the interpretation of my past work.

l.27/28: This was posed by Lelieveld and Crutzen (1990), as $HO_2$ transfers to the aqueous phase, so that gas phase ozone formation through $NO+HO_2$ ceases and dissolved $HO_2$ (through superoxide) reacts with ozone, effectively turning $O_3$ production into $O_3$ loss. To a lesser degree this also applies to $RO_2$.

l.40-43, and l.480: Lelieveld and Crutzen (1990) concluded that net $O_3$ production at particular locations, being subject to cloud processing, can be reduced by 40% (comparable to your CAABA results). Liang and Jacob (1997) referred to the troposphere in the tropics and midlatitudes. On l.480 you are doing the same, although we did not predict a 40% global ozone reduction. Comparing the black and red (ScJAMOC) curves in the lower right panel of Fig. 2, $O_3$ production appears to be strongly reduced indeed. Even the results for ScSTa in Fig. 2 show a substantial reduction in $O_3$ production. Further, Lelieveld and Crutzen (1990) introduced the effects of NOx decrease through nighttime heterogeneous loss of $N_2O_5$ on cloud droplets. A few years later it was shown that $N_2O_5$ is also significantly removed by aqueous aerosols, which moderates the impact of clouds on $N_2O_5$, NOx and oxidants predicted by us in 1990.

References

H. Tost et al. (2006) https://acp.copernicus.org/articles/6/565/2006/

R. Sander et al. (2011) https://gmd.copernicus.org/articles/4/373/2011/

D. Taraborrelli et al. (2009) https://acp.copernicus.org/articles/9/2751/2009/

P. Jöckel et al. (2010) https://gmd.copernicus.org/articles/3/717/2010/

R. Sander et al. (2019) https://gmd.copernicus.org/articles/12/1365/2019/
* * *

---

## Referee Comment (RC2) · Hartmut Herrmann (Referee) · 18 Jan 2021

This submission reminds me of discussions which happened in the very early 1990s on the role of liquid water clouds on tropospheric ozone. Somehow, I thought, atmospheric multiphase chemistry moved on in the 30 years since then but this can hardly be seen fm the perspective of this submission. It would be great to widen the scope here at least to a certain extent.

Wouldn't it be useful to compare the cloud effects on ozone for a simpler and a more advanced chemical aqueous phase scheme eveen beyond the 150 rxn schemes for which Tost et al. (2007) and Jöckel et al. (2016) are cited ? Jöckel et al cites SCAV

(Tost et al., 2006a, 2007a, 2010) - here we turn in a circle, however incomplete as Tost et al 2010 is not mentioned by Rosanka et al. Why not ?

I would really appreciate a reactions table with all reactions and rate constants which are being used and their respective sources.

What developments have been seen in the field which are not implemented in the applied aqueous chemistry schemes and, if so, why not ?

What if the limitation in the aqueous mechanism on small compounds artificially changes the effects expected to be observed with a broader arsenal of cloudwater organics ? Would the findings of the paper still hold ?

I rate the switching of of the Fenton reaction as fatal. I cannot understand why this is done. Fenton is one of the most important OH sources. As a judgement of the aqueous scheme: How do your OH / HO2 concentrations compare to state-of-the-art models and measurements ? The paper tells how Fe valuse could be assessed and there are other ways on top of this - this has been done already.

I appreciate the comments of Jos Lelieveld on the earlier days of aqueous phase modelling. It would be desirable to include all available information to give an accurate look back.

Overall, I feel the paper needs more work.

---

## Author Comment (AC1) · 1 Mar 2021

**Reply to comments of Anonymous Referee #1**

The manuscript presents a revisit to the aqueous chemistry of the superoxide anionin the atmosphere, using both a box and a global model. The focus of the study is on the impact of this newly implemented chemistry on VOCs, OVOCs, HOx, and ozone. Comparisons of the model against satellite methanol and ozone observations are shown, with updates decreasing EMAC's positive ozone bias. Overall the authors clearly demonstrate the importance of including this chemistry in global chemistry models. The science is generally presented in a clear and appropriate way and the manuscript as whole fits the remit of ACP. I would encourage publication effectively as is.

We are very grateful for this positive feedback and for seeing the potential of our contribution to the community. Please find in black the original comments and in red our replies.

**Specific comments**

Line 44 – A suggestion for flow of reading: Add "However" to the beginning of the sentence below.

"By not considering additional in-cloud HO2(aq) sources, Liang and Jacob (1997) underestimated O2(aq) concentrations dampening the in-cloud destruction of O3(aq)."

Done.

Line $\sim$ 175 – Please add a table to section 2.4 to make it easier for the reader to quickly decode the simulation acronyms used elsewhere in the text.

Thank you for this helpful comment. We added a table summarising the characteristics of the gas- and aqueous-phase mechanism used in each simulation performed, using CAABA and EMAC in this study. A reference to it has been added to Sect. 2.4.

Figure 2 caption – Consider moving expansions of families (e.g. VOCs) here and elsewhere to a table in the supplement to make the text more readable.

This is a good point. Removing the definition from the captions will make them easier to read. We think that creating a supplemental material just for this one line does not fit. Thus, we therefore created Appendix A, which includes the definition of $\sum$OVOCs. We also added references to the definition in Appendix A to Sect. 4.1.

Table 4 – Please add some reference numbers from a multi-model study such as TOAR as a column to Table 4. This enables the reader to put these numbers in context (e.g. Loss via bromine seems quite high in this model).

In the revised manuscript, we added two additional comparisons to Table 4 (now Table 5). The multi-model study from TOAR only includes the total chemical production and loss, the dry deposition, the Stratospheric-Tropospheric Exchange, and the burden. Therefore, we added an additional detailed multi-model comparison based on Sherwen et al. (2016), Hu et al. (2017), and Griffiths et al. (2020).

Table 3 and 2 – As with Table 4, is it possible to provide some context for the numbers to another model study? Few will know where these numbers are high or low without context.

In the revised manuscript, we added the detailed gas-phase OH budget from Lelieveld et al. (2016) to Table 2 (now Table 3). We are not aware of any detailed $HO_2$ budget from the literature. Therefore, we couldn't add any comparison to Table 3 (now Table 4).

Please expand all abbreviations/acronyms in table/figure captions or at least link to a table of these (e.g. "Scm" in Table 1).

We now refer to the newly created table summarising the characteristic of each simulation in the caption of all figures and tables.

**References**

Griffiths, P. T., Keeble, J., Shin, Y. M., Abraham, N. L., Archibald, A. T., and Pyle, J. A.: On the Changing Role of the Stratosphere on the Tropospheric Ozone Budget: 1979–2010, Geophysical

Research Letters, 47, e2019GL086 901, https://doi.org/doi.org/10.1029/2019GL086901, 2020.

Hu, L., Jacob, D. J., Liu, X., Zhang, Y., Zhang, L., Kim, P. S., Sulprizio, M. P., and Yantosca, R. M.: Global budget of tropospheric ozone: Evaluating recent model advances with satellite (OMI), aircraft (IAGOS), and ozonesonde observations, Atmospheric Environment, 167, 323–334, https://doi.org/doi.org/10.1016/j.atmosenv.2017.08.036, 2017.

Lelieveld, J., Gromov, S., Pozzer, A., and Taraborrelli, D.: Global tropospheric hydroxyl distribution, budget and reactivity, Atmospheric Chemistry and Physics, 16, 12 477–12 493, https://doi.org/10.5194/acp-16-12477-2016, 2016.

Liang, J. and Jacob, D. J.: Effect of aqueous phase cloud chemistry on tropospheric ozone, Journal of Geophysical Research: Atmospheres, 102, 5993–6001, https://doi.org/10.1029/96JD02957, 1997.

Sherwen, T., Schmidt, J. A., Evans, M. J., Carpenter, L. J., Großmann, K., Eastham, S. D., Jacob, D. J., Dix, B., Koenig, T. K., Sinreich, R., Ortega, I., Volkamer, R., Saiz-Lopez, A., Prados-Roman, C., Mahajan, A. S., and Ordóñez, C.: Global impacts of tropospheric halogens (Cl, Br, I) on oxidants and composition in GEOS-Chem, Atmospheric Chemistry and Physics, 16, 12 239–12 271, https://doi.org/10.5194/acp-16-12239-2016, 2016.

---

## Author Comment (AC2) · 1 Mar 2021

**Reply to the short comments of Johannes Lelieveld**

Compliments for the excellent article that represents a major step forward in the discussion of cloud chemical effects on tropospheric composition. The development of the JAMOC scheme, accounting for comprehensive VOC chemistry, and the successful implementation in the EMAC model is an important accomplishment. The use of JAMOC brings the model significantly closer to observations of VOCs and ozone. Impressive. The results on VOCs and OVOCs, notably of aqueous phase chemistry and considering that most clouds evaporate rather than precipitate, will also offer new angles of approach in studies of organic aerosols.

It should be mentioned that this work was possible as it could build on the EMAC modelling framework, being the effort of a team (of which I am happy to be a member). It has set the stage for comprehensive, global atmospheric chemistry modelling, including the explicit and comprehensive account of VOCs and multiphase processes (e.g. Tost et al., 2007; Taraborrelli et al., 2009; Sander et al., 2011, 2019; Jöckel et al., 2010). I hope the article will be accepted for publication in ACP, while having a few minor comments in view of the interpretation of my past work.

We are very grateful for this positive feedback and for seeing the potential of our contribution to the community. We are fully aware that our work builds on the works from contributors within and outside the EMAC community. Please find in black the original comments and in red our replies.

l.27/28: This was posed by Lelieveld and Crutzen (1990), as HO2 transfers to the aqueous phase, so that gas phase ozone formation through NO+HO2 ceases and dissolved HO2 (through superoxide) reacts with ozone, effectively turning O3 production into O3 loss. To a lesser degree this also applies to RO2.

Thank you for spotting this. We added the appropriate reference to the revised manuscript.

l.40-43, and l.480: Lelieveld and Crutzen (1990) concluded that net O3 production at particular locations, being subject to cloud processing, can be reduced by 40 % (comparable to your CAABA results). Liang and Jacob (1997) referred to the troposphere in the tropics and midlatitudes. On l.480 you are doing the same, although we did not predict a 40 % global ozone reduction. Comparing the black and red (ScJAMOC) curves in the lower right panel of Fig. 2, O3 production appears to be strongly reduced indeed. Even the results for ScSTa in Fig. 2 show a substantial reduction in O3 production. Further, Lelieveld and Crutzen (1990) introduced the effects of NOx decrease through nighttime heterogeneous loss of N2O5 on cloud droplets. A few years later it was shown that N2O5 is also significantly removed by aqueous aerosols, which moderates the impact of clouds on N2O5, NOx and oxidants predicted by us in 1990.

We agree that our last statement might be misleading. Therefore, we specified our statements in line 40-43 and removed parts of the statement in line 480. It now reads: "The predicted $O_3$ loss by clouds is significantly higher than the global estimates by Liang and Jacob (1997) and regional changes might be in the same order of magnitude as predicted by Lelieveld and Crutzen (1990)."

**References**

Jöckel, P., Kerkweg, A., Pozzer, A., Sander, R., Tost, H., Riede, H., Baumgaertner, A., Gromov, S., and Kern, B.: Development cycle 2 of the Modular Earth Submodel System (MESSy2), Geoscientific Model Development, 3, 717–752, https://doi.org/10.5194/gmd-3-717-2010, 2010.

Lelieveld, J. and Crutzen, P. J.: Influences of cloud photochemical processes on tropospheric ozone, Nature, 343, 227–233, https://doi.org/10.1038/343227a0, URL https://doi.org/10.1038/343227a0, 1990.

Liang, J. and Jacob, D. J.: Effect of aqueous phase cloud chemistry on tropospheric ozone, Journal of Geophysical Research: Atmospheres, 102, 5993–6001, https://doi.org/10.1029/96JD02957, 1997.

Sander, R., Baumgaertner, A., Gromov, S., Harder, H., Jöckel, P., Kerkweg, A., Kubistin, D., Regelin, E., Riede, H., Sandu, A., Taraborrelli, D., Tost, H., and Xie, Z.-Q.: The atmospheric chemistry

box model CAABA/MECCA-3.0, Geoscientific Model Development, 4, 373–380, https://doi.org/10.5194/gmd-4-373-2011, 2011.

Sander, R., Baumgaertner, A., Cabrera-Perez, D., Frank, F., Gromov, S., Grooß, J.-U., Harder, H., Huijnen, V., Jöckel, P., Karydis, V. A., Niemeyer, K. E., Pozzer, A., Riede, H., Schultz, M. G., Taraborrelli, D., and Tauer, S.: The community atmospheric chemistry box model CAABA/MECCA-4.0, Geoscientific Model Development, 12, 1365–1385, https://doi.org/10.5194/gmd-12-1365-2019, 2019.

Taraborrelli, D., Lawrence, M. G., Butler, T. M., Sander, R., and Lelieveld, J.: Mainz Isoprene Mechanism 2 (MIM2): an isoprene oxidation mechanism for regional and global atmospheric modelling, Atmospheric Chemistry and Physics, 9, 2751–2777, https://doi.org/10.5194/acp-9-2751-2009, 2009.

Tost, H., Jöckel, P., Kerkweg, A., Pozzer, A., Sander, R., and Lelieveld, J.: Global cloud and precipitation chemistry and wet deposition: tropospheric model simulations with ECHAM5/MESSy1, Atmospheric Chemistry and Physics, 7, 2733–2757, https://doi.org/10.5194/acp-7-2733-2007, 2007.

---

## Author Comment (AC3) · 1 Mar 2021

**Reply to comments of Hartmut Herrmann (Referee #2)**

This submission reminds me of discussions which happened in the very early 1990s on the role of liquid water clouds on tropospheric ozone. Somehow, I thought, atmospheric multiphase chemistry moved on in the 30 years since then but this can hardly be seen fm the perspective of this submission. It would be great to widen the scope here at least to a certain extent.

Thank you very much for the helpful comments. Please find in black the original comments and in red our replies.

Wouldn't it be useful to compare the cloud effects on ozone for a simpler and a more advanced chemical aqueous phase scheme eveen beyond the 150 rxn schemes for which Tost et al. (2007) and Jöckel et al. (2016) are cited ? Jöckel et al cites SCAV Tost et al. (2006, 2007, 2010) - here we turn in a circle, however incomplete as Tost et al. (2010) is not mentioned by Rosanka et al. Why not?

We agree with the reviewer that a comparison between a simple and advanced chemical aqueous-phase mechanisms is needed and therefore forms the basis of this manuscript. A detailed in-cloud oxidation scheme, which is suitable for global model applications, was not available for our study. Therefore, we developed the detailed in-cloud oxidation scheme JAMOC in our companion paper within the box model CAABA and apply it on a global scale using EMAC in this paper. When JAMOC is applied within EMAC, it represents the phase transfer of more than 350 species, includes 43 equilibria, and 289 photo-oxidation reactions. In the revised manuscript, we adjusted the description of the different aqueous-phase mechanisms used (Sect. 2.1.1.), in order to further illustrate their differences. In addition, a new table has been added (now Table 1), which summarises the characteristics of the different gas- and aqueous-phase mechanisms used. The table also includes the number of species that are partitioned into cloud droplets, the number of equilibria, and photo-oxidation reactions.

Tost et al. (2010) investigate the influence of convection parameterisations on the chemical composition. Here, the different schemes result in varying precipitation patterns and varying vertical distributions of cloud and precipitable water. This ultimately influences the scavenging efficiency. Since the same convection parameterisation is used across all EMAC simulations, the relative change is not directly influenced by this. Still, the magnitude of the changes might differ with another convection parameterisation used. However, investigating the influence of varying convection parameterisation on the total magnitude is outside the scope of this study.

I would really appreciate a reactions table with all reactions and rate constants which are being used and their respective sources.

We fully agree that complete information about the reaction mechanism, rate constants, and references is necessary for a thorough review. We have already published this, as indicated at the bottom of page 2 in the originally submitted version: "For the detailed in-cloud OVOC oxidation scheme, the Jülich Aqueous-phase Mechanism of Organic Chemistry (JAMOC) suitable for global model applications is developed and implemented into the atmospheric chemistry mechanism Module Efficiently Calculating the Chemistry of the Atmosphere (MECCA) in our companion paper by Rosanka et al. (2020a)." In the revised version of the manuscript, we now include an additional table (now Table 1) that summarises the characteristics of the gas- and aqueous-phase chemical mechanism used in each simulation. For the table of the reactions used in the simple (Scm) and EMAC's standard mechanism (ScSTA), the reader may find the mechanism tables in the listed references.

What developments have been seen in the field which are not implemented in the applied aqueous chemistry schemes and, if so, why not ?

We agree with the referee that understanding the recent developments and missing representations in the newly developed mechanism (JAMOC) is very important in order to understand the limitations of the simulations performed. Therefore, we included a detailed description on the mechanism development and limitations in our companion model description paper by Rosanka et al. (2020). This also includes a short review of the recent advancements. Repeating this information in this manuscript does not fit the idea of companion papers. Therefore, please refer to our companion paper for the mechanism limitations.

What if the limitation in the aqueous mechanism on small compounds artificially changes the effects

expected to be observed with a broader arsenal of cloudwater organics? Would the findings of the paper still hold ?

We expect that even if a broader set of organics was transferred into cloud droplets, they would still react with OH, leading to the formation of $HO_2$. If the number of organics treated in JAMOC was increased and if we assume that each of the organics reacts with OH with a rate coefficient of $k_{C,OH} = (3.8 \pm 1.9) \times 10^8$ $M^{-1}$ $s^{-1}$ (Arakaki et al., 2013), the predicted in-cloud OH concentration would be different. Therefore, we expect that the magnitude of the predicted change for each individual organic could vary but the overall tendency would stay the same. At the same time, the increased formation of $HO_2$ would increase the in-cloud destruction of $O_3$. In the future, we plan to expand JAMOC to include the oxidation of additional organic compounds.

I rate the switching of of the Fenton reaction as fatal. I cannot understand why this is done. Fenton is one of the most important OH sources. As a judgement of the aqueous scheme: How do your OH/HO2 concentrations compare to state-of-the-art models and measurements ? The paper tells how Fe valuse could be assessed and there are other ways on top of this - this has been done already.

We understand the concerns of the referee that the missing Fenton's chemistry will influence the resulting impact of JAMOC. In our box model calculations presented in Sect. 3, we predict average in-cloud concentrations of $1.3 \times 10^{-13}$ M and $2.5 \times 10^{-8}$ M for OH and $HO_2$, respectively. These values are very similar to the predictions of the box-model mechanism CLEPS (see Fig. 4 in Mouchel-Vallon et al., 2017) and observations and predictions by Tilgner et al. (2013) and Arakaki et al. (2013). We expect that the missing Fenton's chemistry will therefore mainly influence the resulting impact in magnitude but the tendency will stay the same. Although we think that the effect of iron emissions in EMAC is not within the scope of this study, this is already work in progress, and we plan to present Fenton's chemistry in EMAC in a future publication. Overall, we intend this paper and our companion paper in GMD (Rosanka et al., 2020) to be a starting point for further aqueous-phase mechanism developments in EMAC. In the revised version of the manuscript, we now include a short analysis of the in-cloud OH and $HO_2$ concentrations in CAABA.

I appreciate the comments of Jos Lelieveld on the earlier days of aqueous phase modelling. It would be desirable to include all available information to give an accurate lookback.

We greatly appreciate the comments of Jos Lelieveld and have implemented them accordingly. We have expanded the introduction of the revised manuscript. It now includes a discussion on the impact of aqueous-phase chemistry on the formation of secondary organic aerosol (SOA). In addition, we now include a review on aqueous-phase mechanisms currently applied in regional and global models.

Overall, I feel the paper needs more work.

Following your comments and the comments of the first referee, we revised the manuscript by adding the following aspects:

- The discussion on the aqueous-phase $O_3$ chemistry was updated following the recommendations of Jos Lelieveld.

- The introduction was expanded to include a review on the importance of aqueous-phase processes on the formation of SOA.

- The aqueous-phase mechanisms available in other global and regional models are reviewed in the introduction.

- We expanded the description of the organics explicitly reacting in JAMOC in Sect. 2.1.1.

- An additional table has been added, which summarises the characteristics of the gas- and aqueous-phase mechanisms used in each simulation.

- Sect. 3 now includes a discussion of the OH and $HO_2$ concentrations predicted within the cloud droplets modelled by CAABA.

- An appendix was added, which includes the definition of $\sum OVOCs$ used in Figs. 2 and 3.

- Table 3 (previously Table 2) now includes a comparison to the gas-phase OH budget presented by Lelieveld et al. (2016).

- The $O_x$ budget presented in Table 5 (previously Table 4) now includes a multi-model comparison and a comparison to the budgets presented in TOAR.

- The IASI $O_3$ retrieval has been improved over deserts (e.g. Sahara) in Figure 11.

**References**

Arakaki, T., Anastasio, C., Kuroki, Y., Nakajima, H., Okada, K., Kotani, Y., Handa, D., Azechi, S., Kimura, T., Tsuhako, A., and Miyagi, Y.: A General Scavenging Rate Constant for Reaction of Hydroxyl Radical with Organic Carbon in Atmospheric Waters, Environmental Science & Technology, 47, 8196–8203, https://doi.org/10.1021/es401927b, pMID: 23822860, 2013.

Jöckel, P., Tost, H., Pozzer, A., Kunze, M., Kirner, O., Brenninkmeijer, C. A. M., Brinkop, S., Cai, D. S., Dyroff, C., Eckstein, J., Frank, F., Garny, H., Gottschaldt, K.-D., Graf, P., Grewe, V., Kerkweg, A., Kern, B., Matthes, S., Mertens, M., Meul, S., Neumaier, M., Nützel, M., Oberländer-Hayn, S., Ruhnke, R., Runde, T., Sander, R., Scharffe, D., and Zahn, A.: Earth System Chemistry integrated Modelling (ESCiMo) with the Modular Earth Submodel System (MESSy) version 2.51, Geoscientific Model Development, 9, 1153–1200, https://doi.org/10.5194/gmd-9-1153-2016, 2016.

Lelieveld, J., Gromov, S., Pozzer, A., and Taraborrelli, D.: Global tropospheric hydroxyl distribution, budget and reactivity, Atmospheric Chemistry and Physics, 16, 12 477–12 493, https://doi.org/10.5194/acp-16-12477-2016, 2016.

Mouchel-Vallon, C., Deguillaume, L., Monod, A., Perroux, H., Rose, C., Ghigo, G., Long, Y., Leriche, M., Aumont, B., Patryl, L., Armand, P., and Chaumerliac, N.: CLEPS 1.0: A new protocol for cloud aqueous phase oxidation of VOC mechanisms, Geoscientific Model Development, 10, 1339–1362, https://doi.org/10.5194/gmd-10-1339-2017, 2017.

Rosanka, S., Sander, R., Wahner, A., and Taraborrelli, D.: Oxidation of low-molecular weight organic compounds in cloud droplets: development of the JAMOC chemical mechanism in CAABA/MECCA (version 4.5.0gmdd), Geoscientific Model Development Discussions, 2020, 1–18, https://doi.org/10.5194/gmd-2020-337, companion paper, 2020.

Tilgner, A., Bräuer, P., Wolke, R., and Herrmann, H.: Modelling multiphase chemistry in deliquescent aerosols and clouds using CAPRAM3.0i, Journal of Atmospheric Chemistry, 70, 221–256, https://doi.org/10.1007/s10874-013-9267-4, 2013.

Tost, H., Jöckel, P., Kerkweg, A., Sander, R., and Lelieveld, J.: Technical note: A new comprehensive SCAVenging submodel for global atmospheric chemistry modelling, Atmospheric Chemistry and Physics, 6, 565–574, https://doi.org/10.5194/acp-6-565-2006, 2006.

Tost, H., Jöckel, P., Kerkweg, A., Pozzer, A., Sander, R., and Lelieveld, J.: Global cloud and precipitation chemistry and wet deposition: tropospheric model simulations with ECHAM5/MESSy1, Atmospheric Chemistry and Physics, 7, 2733–2757, https://doi.org/10.5194/acp-7-2733-2007, 2007.

Tost, H., Lawrence, M. G., Brühl, C., Jöckel, P., Team, T. G., and Team, T. S.-O.-D.: Uncertainties in atmospheric chemistry modelling due to convection parameterisations and subsequent scavenging, Atmospheric Chemistry and Physics, 10, 1931–1951, https://doi.org/10.5194/acp-10-1931-2010, 2010.